# Divacancy and resonance level enables high thermoelectric performance in *n*-type SnSe polycrystals

Yaru Gong[1,6], Wei Dou[1,6], Bochen Lu[2,6], Xuemei Zhang[3], He Zhu[1], Pan Ying ●[1], Qingtang Zhang[1], Yuqi Liu[1], Yanan Li[1], Xinqi Huang[1], Muhammad Faisal Iqbal[1], Shihua Zhang[1], Di Li[4], Yongsheng Zhang ●[5] ✉, Haijun Wu ●[2] ✉ & Guodong Tang ●[1] ✉

*N*-type polycrystalline SnSe is considered as a highly promising candidates for thermoelectric applications due to facile processing, machinability, and scalability. However, existing efforts do not enable a peak *ZT* value exceeding 2.0 in *n*-type polycrystalline SnSe. Here, we realized a significant *ZT* enhancement by leveraging the synergistic effects of divacancy defect and introducing resonance level into the conduction band. The resonance level and increased density of states resulting from tungsten boost the Seebeck coefficient. The combination of the enhanced electrical conductivity (achieved by increasing carrier concentration through $WCl_6$ doping and Se vacancies) and large Seebeck coefficient lead to a high power factor. Microstructural analyses reveal that the co-existence of divacancy defects (Se vacancies and Sn vacancies) and endotaxial W- and Cl-rich nanoprecipitates scatter phonons effectively, resulting in ultralow lattice conductivity. Ultimately, a record-high peak *ZT* of 2.2 at 773 K is achieved in *n*-type $SnSe_{0.92}$ + 0.03$WCl_6$.

Thermoelectric material semiconductors can enable direct interconversion between electricity and waste heat, which have exhibited considerable potential to tackle energy and environmental crisis[1–4]. The thermoelectric performance is usually evaluated by the dimensionless figure of merit ($ZT = S^2\sigma T/\kappa_T$), where $S$ is the Seebeck coefficient, $\sigma$ is the electrical conductivity, $T$ is the absolute temperature, and $\kappa_T$ is the total thermal conductivity[1–4]. The electrical transport properties can be expressed in terms of the power factor ($PF = S^2\sigma$). The total thermal conductivity ($\kappa_T$) is mainly composed of two parts: the electronic ($\kappa_e$) and the lattice ($\kappa_L$) thermal contributions. Optimizing the power factor and simultaneous lowering the lattice thermal conductivity are critical for advanced thermoelectric materials. In the regulation of electrical properties, the band convergence[5–7], resonance level[8,9], carrier concentration optimization[10,11], and energy filtering effects[12,13] can effectively enhance the power factor of materials. In addition, nanostructuring[14–16], phase separation[17,18], and all-scale hierarchical architecturing[19,20] are widely used to modulate the phonon transport process in materials and suppress the heat transport.

Due to its low toxicity, earth abundant elements, and high thermoelectric properties, the tin selenide (SnSe) compound attracts widespread attention and emerges as an enormous potential for medium-temperature thermoelectric applications[1,4,21]. SnSe adopts an

[1]National Key Laboratory of Advanced Casting Technologies, MIIT Key Laboratory of Advanced Metallic and Intermetallic Materials Technology, Engineering Research Center of Materials Behavior and Design, Ministry of Education, Nanjing University of Science and Technology, Nanjing, China. [2]State Key Laboratory for Mechanical Behavior of Materials, Xi'an Jiaotong University, Xi'an, China. [3]School of Physics and Electronic Information Engineering, Engineering Research Center of Nanostructure and Functional Materials, Ningxia Normal University, Guyuan, Ningxia, China. [4]Key Laboratory of Materials Physics, Institute of Solid State Physics, Chinese Academy of Sciences, Hefei, China. [5]Advanced Research Institute of Multidisciplinary Sciences, Qufu Normal University, Qufu, Shandong Province, China. [6]These authors contributed equally: Yaru Gong, Wei Dou, Bochen Lu. ✉e-mail: yshzhang@theory.issp.ac.cn; wuhaijunnavy@xjtu.edu.cn; tangguodong@njust.edu.cn

orthorhombic layer crystal structure (*Pnma*) at room temperature, which derives from three-dimensional distortion of NaCl structure[22]. SnSe undergoes a displacive (shear) phase transition from *Pnma* to *Cmcm* structure around 800 K[23,24]. It has a combination of strong intralayer (the strong Sn−Se covalent bonds) and weak interlayer (the weak van der Walls interactions) forces. Strong lattice anharmonicity in SnSe gives rise to its low lattice thermal conductivity[22]. Outstanding *ZT* of ~2.6 at 923 K and ~2.8 at 773 K have been well reported in *p*-type[22] and *n*-type[25] SnSe single crystals, respectively. However, suffered from the laborious and rigid synthesis procedure, high cost for production, and poor mechanical properties, SnSe single crystals are undesirable for assembling practical devices for commercialization[4,21]. On the other hand, the polycrystalline SnSe compounds have advantage on facile processing, machinability and scalability. On these accounts, it is highly desirable to developing polycrystalline SnSe with *ZT* values comparable to those of SnSe crystals owing to its facile processing, machinability, and scale-up application. To date, many p-type polycrystalline SnSe with excellent performance have been developed through traditional chemical doping (Na, K, Cd)[26–29], phase separation[18,30], introducing Sn vacancies[31], and oxide removing strategy[32,33]. In order to match with the *p*-type counterpart to form efficient thermoelectric devices, the development of high-performance n-type polycrystalline SnSe materials is vital for practical applications[4,21,34]. Donor doping (Br, BiCl₃, CeCl₃, Bi, Sb, Re)[35–41], and band gap engineering[42], WSe₂/SnSe p-n junctions[43] and MoCl₅[44] have been adopted to raise *ZT* of *n*-type SnSe polycrystalline. Even with a synergic approach involving strategies above, enormous efforts do not enable its peak *ZT* value exceeding 2.0, much more inferior than those of SnSe crystals and *p*-type polycrystalline. This is due to the high thermal conductivity and lower power factor in the n-type polycrystalline. SnSe always exhibit a *p*-type semiconductor behavior due to its intrinsic Sn vacancies. The creation of Se vacancies can increase the electron concentration and realize the desired n-type polycrystalline SnSe system. Moreover, vacancies, which disrupt translation symmetry, can act as phonon scattering centers to suppress the lattice thermal conductivity. According to the Pisarenko relationship, the carrier concentrations increase leads to a significantly reduced Seebeck coefficient of doped *n*-type polycrystalline SnSe system, which greatly limits the ability for further optimizing the electrical transport properties and *ZT* value. To overcome the Seebeck coefficient decrease under carrier optimization, creating resonant levels is a fascinating route to boost the Seebeck coefficient and power factor.

Here, a new strategy of synergy of divacancy defects and doping resonant state was demonstrated to an effective route for obtaining high thermoelectric performance of *n*-type polycrystalline SnSe (Fig. 1a). We found that the divacancy defects and endotaxial W- and Cl-rich nanoprecipitates construct a multiple-scale microstructure, resulting in strong phonon scatterings and thus an ultralow lattice thermal conductivity. Electronically, it is revealed that the resonant levels and increase DOS in the electronic structure of SnSe caused by W dopant produce remarkable enhancement of Seebeck coefficient. In the meanwhile, the creation of Se vacancies and WCl₆ doping give rise to increase of electron concentration, which help to achieve high power factor in *n*-type polycrystalline SnSe₀.₉₂ + 0.03WCl₆. Consequently, we successfully achieved a record-high *ZT* value of 2.2 in *n*-type polycrystalline SnSe₀.₉₂ + 0.03WCl₆, which outperforms the most state-of-the-art *n*-type thermoelectric systems (Fig. 1b), highlighting the prospect of advancing thermoelectrics[45–51].

## Results and discussion
### Crystal structure and phase description
In Fig. 2a, the X-ray diffraction (XRD) peaks of all SnSe₀.₉₂ + x WCl₆ (x = 0, 0.01, 0.02, 0.03, 0.04) powders are well indexed with the diffraction peaks of the *Pnma* orthorhombic SnSe structure, and no impurity phase is detected. Meanwhile, the (011), (111), and (400) peaks shift to higher angle with the increase of WCl₆ content, indicating that the introduction of W and Cl will lead to the shrinking of the SnSe lattice. As shown in Fig. 2b, the Rietveld refinement of XRD were performed to obtain more comprehensive information on the SnSe₀.₉₂ + 0.03WCl₆ crystal structures. The atom occupied positions and detailed lattice constants after corrections are shown in Supplementary Table 1 and Supplementary Table 2, the lattice parameters of all SnSe₀.₉₂ + x WCl₆ (x = 0.01, 0.02, 0.03, 0.04) are smaller than those of SnSe₀.₉₂. W and Cl substitute Sn and Se and occupy similar Wyckoff positions, respectively, and a certain content of vacancies exists at both Sn and Se positions. It is well-known that SnSe crystallizes in a layered structure with an orthorhombic *Pnma* symmetry, where the layers spreading over *bc*-direction are comprised of covalently bonded Se and Sn atoms to form zigzag chains. For the local Sn−Se building block, each Sn atom is bonded with three Se atoms in a distorted T-shaped geometry. To understand the impact of WCl₆ on the local structure regulation, the synchrotron pair distribution function (PDF) analysis was performed on both SnSe₀.₉₂ + 0.03WCl₆ and SnSe₀.₉₂ samples. Figure 2c illustrates the SnSe structure, where number 1, 2, and 3 denote the nearest neighbor Sn−Se, Sn−Se, and Sn−Sn bonds, respectively. Supplementary Fig. 1 compares the G(r) patterns in the real space (2–30 Å). From the low-r patterns (Fig. 2d), the first peak, which is assigned to the nearest neighboring Sn−Se bonds, shows a slight increase in bond length due to the WCl₆ doping. In addition, the second peak located around 3.2 Å is assigned to the next-nearest Sn… Se distances within the *bc* plane with almost no change. The distance of Sn…Sn pairs along the a-axis direction between layers corresponds to

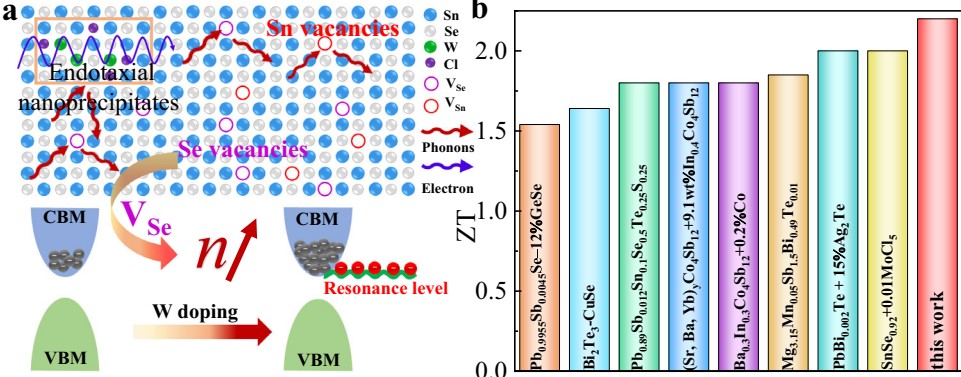

**Fig. 1 | Synergistic optimization of the carrier and phonon transports via divacancy defect and introducing resonance level. a** Modulation mechanism of WCl₆ doping on SnSe electron-phonon transport. **b** Comparisons of *ZT* for SnSe₀.₉₂ + 0.03WCl₆ with *n*-type thermoelectric systems.

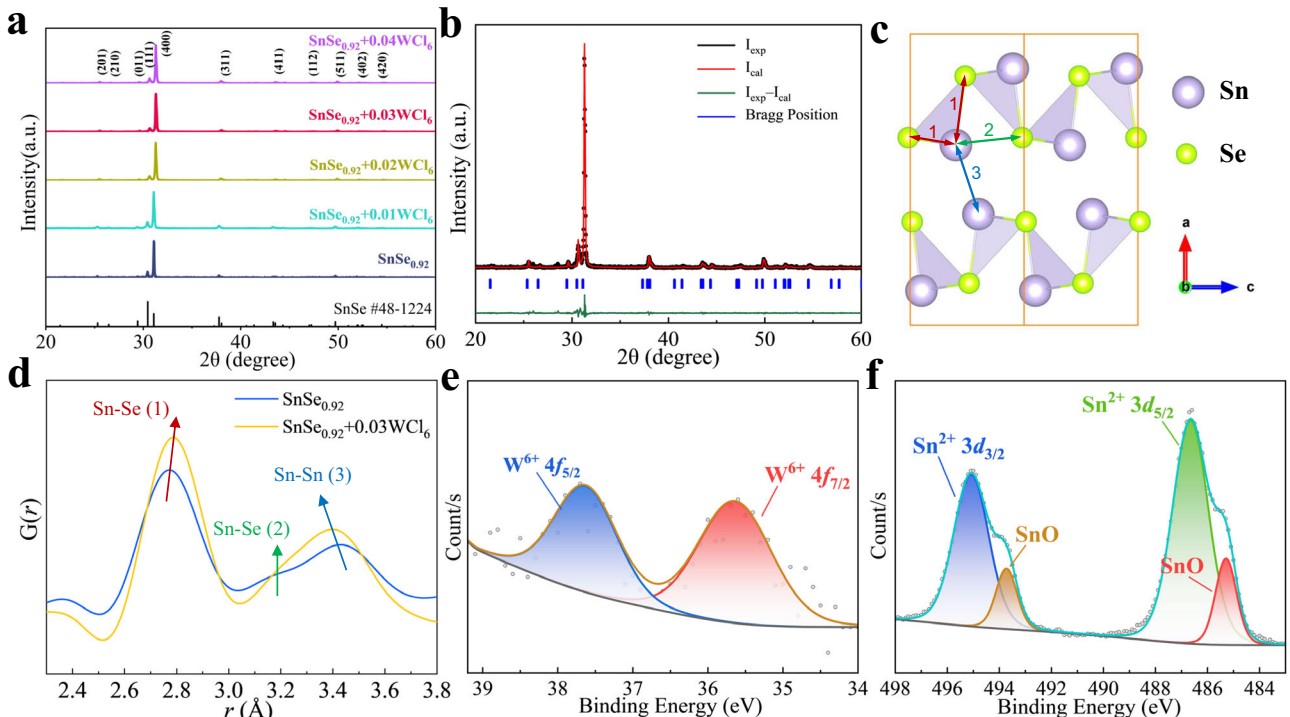

**Fig. 2 | Crystal structure and XPS characterization for SnSe$_{0.92}$ + x WCl$_6$. a** XRD patterns of SnSe$_{0.92}$ + x WCl$_6$. **b** Rietveld refinement details of SnSe$_{0.92}$ + 0.03WCl$_6$. **c** SnSe structure: with purple and green spheres indicate Sn and Se atoms, respectively. **d** The comparison of the G(r) patterns in the low space (2.3–3.8 Å). XPS core-level spectra of **e** W$^{6+}$ 4f, **f** Sn$^{2+}$ 3d peak from SnSe$_{0.92}$ + 0.03WCl$_6$.

the third peak in the G(r) patterns at around 3.5 Å, obviously with a strong contraction of the interlayer spacing connected by van der Waals forces. The synchrotron pair distribution function analysis show the decrease in the lattice parameter $a$ after WCl$_6$ doping, which is consistent with the results of Rietveld refinement. X-ray photoelectron spectroscopy (XPS) on the SnSe$_{0.92}$ + 0.03WCl$_6$ sample was performed to determine the ionic states of elements (Fig. 2e, f). Sn exists in two bonding modes, where peaks with binding energies of (495.1 and 486.7 eV) and (493.7 and 485.3 eV) can be attributed to Sn$^{2+}$ and SnO[52,53]. In addition, the peaks of W can be assigned to W$^{6+}$ with the binding energies of 37.9 and 35.7 eV[54]. The lattice contraction can be ascribed to the smaller ionic radius of W$^{6+}$ (0.6 Å) and Cl$^-$ (1.81 Å) than those of Sn$^{2+}$ (1.18 Å) and Se$^{2-}$ (1.98 Å)[55].

## Electrical transport properties

Figure 3a shows the temperature dependence of electrical conductivity ($\sigma$) of SnSe$_{0.92}$ + x WCl$_6$ (x = 0.01, 0.02, 0.03, 0.04) parallel to the pressure direction. A similar temperature-dependent trend is observed for all samples, displaying typical semiconductor transport behaviors[40,42,44]. SnSe$_{0.92}$ + x WCl$_6$ samples exhibit a significantly enhanced $\sigma$ compared to pristine SnSe$_{0.92}$. The SnSe$_{0.92}$ + 0.01WCl$_6$ sample has the highest $\sigma$ among all samples. The largest $\sigma$ of SnSe$_{0.92}$ + 0.01WCl$_6$ reaches to 39.2 S cm$^{-1}$ at 823 K, which is three times higher than that of SnSe$_{0.92}$ (12.67 S cm$^{-1}$). Then $\sigma$ decreases as the doping concentration further increases. As shown in Supplementary Table 3, the carrier concentration ($n$) increases after WCl$_6$ doping. With 0.01WCl$_6$ doping (SnSe$_{0.92}$ + 0.01WCl$_6$), $n$ increases to 4.174 × 10$^{18}$ cm$^{-3}$ from 1.25 × 10$^{17}$ cm$^{-3}$ for the pristine SnSe$_{0.92}$ sample. In our SnSe samples, 0.08 Se vacancy was designed in the sample composition, which inhibits the formation of large amount of intrinsic Sn vacancies, resulting in an $n$-type conductivity. Simultaneously, the introduction of higher valence W$^{6+}$ and lower valence Cl$^-$ increases the carrier concentration of SnSe$_{0.92}$ + x WCl$_6$. A possible carrier optimization mechanism in SnSe$_{0.92}$ + xWCl$_6$ samples is given as:

$$\text{SnSe}(p-\text{type}) \rightarrow \text{SnSe}_{0.92} + 0.08e^-(n-\text{type}) \tag{1}$$

$$\text{SnSe}_{0.92} + 0.01\text{WCl}_6 \rightarrow \text{Sn(W)}_{1.01}\text{Se(Cl)}_{0.98} + 0.16e^- \tag{2}$$

$$\text{SnSe}_{0.92} + 0.02\text{WCl}_6 \rightarrow \text{Sn(W)}_{1.02}\text{Se(Cl)}_{1.04} + 0.14e^- \tag{3}$$

As a result, the sharp increase in $n$ leads to a significant increase of $\sigma$. The carrier concentration decreases sharply from x = 0.02. The Cl will progressively occupy the Se vacancies with increasing WCl$_6$ doping content. The introduction of Cl dopants results in an imbalance, the density of Sn vacancies is larger than that of Se vacancies, which is evidenced by aberration-corrected STEM analysis. This discrepancy is anticipated to culminate in a decline in carrier concentration. $\sigma$ of high performance SnSe$_{0.92}$ + 3% WCl$_6$ sample maintain very low value from 300 to 475 K. Then $\sigma$ sharply increases with increasing temperature above 475 K. The temperature dependence of carrier concentration ($n$) and carrier mobility ($\mu$) were measured for further understanding the electrical conductivity change with temperature (Supplementary Fig. 2). The carrier concentration of the SnSe$_{0.92}$ + 0.03WCl$_6$ sample increase above 475 K, which resulting from thermal activation[31]. In the meanwhile, the carrier mobility abruptly increases with temperature. Therefore, the notable increase of $\sigma$ above 475 K can be ascribed to both the increased carrier concentration and carrier mobility.

The temperature-dependent Seebeck coefficient ($S$) of SnSe$_{0.92}$ + x WCl$_6$ (x = 0.01, 0.02, 0.03, 0.04) along the pressing direction is shown in Fig. 3b. The negative values of $S$ indicate that SnSe$_{0.92}$ + x WCl$_6$ samples are $n$-type semiconductors. It is found that all WCl$_6$ doped SnSe$_{0.92}$ samples exhibit significantly enhanced |$S$| in the whole temperature range. Sharp increase of |$S$| can be observed in

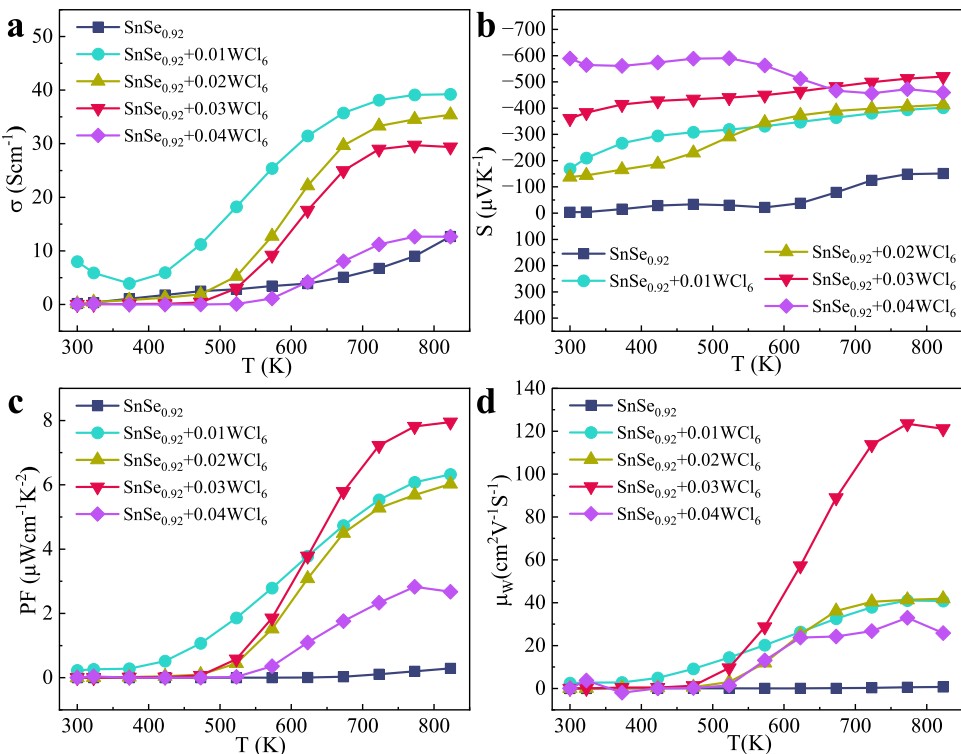

**Fig. 3 | The temperature dependence of electrical transport properties of SnSe$_{0.92}$ + x WCl$_6$. a** Electrical conductivity ($\sigma$), **b** Seebeck coefficient ($S$), **c** Power factor (*PF*), **d** weighted mobility ($\mu_W$) along the pressing direction.

both SnSe$_{0.92}$ + 0.03WCl$_6$ and SnSe$_{0.92}$ + 0.04WCl$_6$. The high-performance SnSe$_{0.92}$ + 0.03WCl$_6$ sample exhibits the relatively high Seebeck coefficients over full temperature range compared to the reported utmost *n*-type SnSe-based materials (Supplementary Fig. 3)[40–42 44,56,57]. The Seebeck coefficient of SnSe$_{0.92}$ + 0.04WCl$_6$ decreases with increasing temperature, especially at the high-temperature regions. This is consistent with the sharply increased carrier concentration (Supplementary Fig. 2), suggesting bipolar conduction. *n* of SnSe$_{0.92}$ + 0.04WCl$_6$ reaches as high as $1.25 \times 10^{19}$ cm$^{-3}$ at 823 K, which is about two orders of magnitude larger than that of $1.3 \times 10^{17}$ cm$^{-3}$ at 510 K. The bipolar behavior is consistent with our band structure calculations (Detailed in Supplementary Fig. 4), which indicates the substantial reduction in the band gap due to high W doping. Therefore, a bipolar conduction process is expected with rising temperature. To further investigate the mechanism of large enhancement of Seebeck coefficient of WCl$_6$ doped SnSe$_{0.92}$, the Pisarenko relation between |S| and the carrier concentration ($n_H$) is calculated based on the single parabolic band (SPB) model, as illustrated in the solid line of Supplementary Fig. 5. This relationship can give valuable hints on the changes in electronic structures. Similar to the reported *n*-type SnSe using band engineering[42,58], the data of SnSe$_{0.92}$ + 0.04WCl$_6$ locate above the Pisarenko curve, clearly predicting that the introduction of WCl$_6$ modifies the electronic band structures of SnSe.

To elucidate the origin of the enhanced electrical transport properties, the electronic band structure have been calculated by DFT calculations. From our theoretical calculations, the lattice parameters of pristine SnSe are $a = 11.81$, $b = 4.22$, and $c = 4.53$ Å, which are in good agreement with the experimental results ($a = 11.58$, $b = 4.22$, and $c = 4.40$ Å)[22]. Its band structures (Fig. 4a) show an indirect band gap of 0.61 eV. Focusing on the conduction bands (the *n*-type properties), the conduction band minimum (CBM1) is along the Γ−X direction. In addition, the second (CBM2) and third (CBM3) conduction band minima site at the Γ point and along the Z−U direction, respectively. The energy differences from the CBM1 to CBM2 ($\Delta E^{CBM1-CBM2}$) and to

CBM3 ($\Delta E^{CBM1-CBM3}$) are 0.13 and 0.48 eV, respectively (Fig. 4a and Table 1). To elucidate the effects of defects (Se vacancy and W doping) on the properties of SnSe, we carry out the electronic structures (Fig. 4) of these defect systems (SnSe$_{0.92}$ and Sn$_{0.963}$W$_{0.027}$Se$_{0.92}$, Supplementary Fig. 6). We notice that once introducing 8% of Se vacancy in SnSe (SnSe$_{0.92}$, Supplementary Fig. 6c), the crystal structure undergoes the local geometry distortion (Supplementary Fig. 6) and modifies the electronic structures. The Se vacancies induce the slight decrease of band energy differences of CBM1, CBM2 and CBM3 ($\Delta E^{CBM1-CBM2} = 0.11$ eV and $\Delta E^{CBM1-CBM3} = 0.30$ eV, Fig. 4b and Table 1), indicating the band convergency and increase density of states (DOS) around the CBM (Fig. 4c). This will increase the DOS effective mass and the Seebeck coefficient. Below the Fermi energy level, a small DOS peak is also found, which is from the Sn-p energy level (Supplementary Fig. 7b). However, since it is below the Fermi level and far away from the conduction band minimum, it is a defect energy level in the band gap and would not contribute to the *n*-type electron transport. On the other hand, the band gap with Se vacancy is decreased to 0.45 eV, which might increase the possible bipolar effect. With further W doping (Sn$_{0.963}$W$_{0.027}$Se$_{0.92}$, Fig. 4d), the substituted W atom not only binds with the neighboring Se atoms within the Sn−Se layer (the W−Se bond lengths are 2.48 and 2.55 Å), but also binds with a Se across the layer (the bond length is 2.53 Å). For the pristine SnSe compound, due to the layered structure, the interlayer electrical conductivity is very low[1,21,22]. Thus, the total electrical conductivity of pristine SnSe polycrystals is also low (Fig. 3a). However, the W dopant in SnSe can bridge two Sn−Se layers (Fig. 4d). The interlayer bonds will facilitate the electron transport across the layers, and boost the electrical conductivity. This is consistent with the experimentally measured increased electrical conductivity of W doping compared to those of the pristine SnSe (Fig. 3a). For the electronic structures (Fig. 4e, f), the W doping recovers the band gap to 0.6 eV and benefits for the further band convergency: $\Delta E^{CBM1-CBM2} = 0.08$ eV and $\Delta E^{CBM1-CBM3} = 0.24$ eV (Table 1). This band convergency clearly induces the DOS increasement around

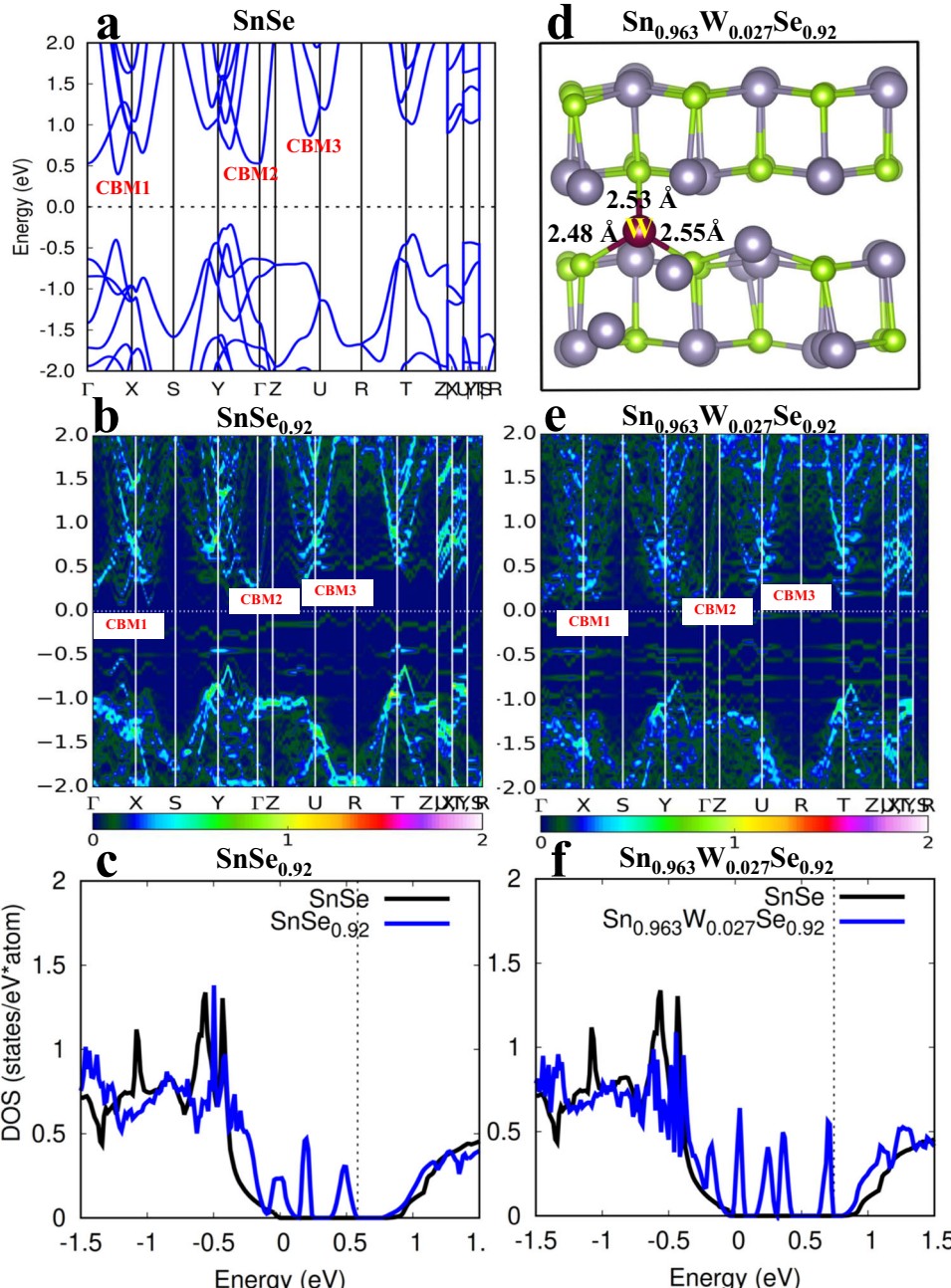

**Fig. 4 | Electronic band structures. a** Electronic band structures of SnSe. **b** Crystal structures of $Sn_{0.963}W_{0.028}Se_{0.92}$. Unfolding band structures of **c** $SnSe_{0.92}$ and **d** $Sn_{0.963}W_{0.027}Se_{0.92}$. The scale bar is the magnitude of the spectral weight, which characterizes the probability of the primitive cell eigenstates contributing to a particular supercell eigenstate of the same energy. Density of states of **e** $SnSe_{0.92}$ and **f** $Sn_{0.963}W_{0.027}Se_{0.92}$. The dashed line represents the Fermi energy level.

the CBM (Fig. 4f). From the DOS of W-doped SnSe (Fig. 4f), we find a strong peak at the Fermi level as well. Unlike the defect energy level in the DOS of $SnSe_{0.92}$, the formation of W–Se bonds in W doped SnSe indicates that the strong interactions between W and the SnSe matrix. This will modify the corresponding electronic structures of SnSe matrix and form the resonant energy level just below the conduction band minimum (CBM) and more DOS peaks below the Fermi level (Fig. 4e, f). Actually, this resonant DOS peak is from the d energy level of W dopant (Supplementary Fig. 7c). This W induced DOS peak will involve in the *n*-type electron transport. Moreover, the introduction of W dopant leads to the emergence of additional DOS peaks below the Fermi level means that W doped $SnSe_{0.92}$ has a higher electron carrier concentration than $SnSe_{0.92}$. The increased DOS (or the induced

resonant energy level around the CBM), the W–Se bonds bridging SnSe interlayers and the increased carrier concentration will significantly boost the Seebeck coefficient and electrical conductivity of W doped $SnSe_{0.92}$, respectively. These are consistent with the experimental observations.

Figure 3c shows the temperature dependence of power factor (*PF*) of $SnSe_{0.92} + x\ WCl_6$ (x = 0.01, 0.02, 0.03, 0.04) along the pressing direction. All $SnSe_{0.92} + x\ WCl_6$ samples exhibit significantly enhanced *PF* over $SnSe_{0.92}$. The highest *PF* of 7.95 μW cm$^{-1}$ K$^{-2}$ is obtained in $SnSe_{0.92} + 0.03WCl_6$ at 773 K, which is 27 times bigger than that of $SnSe_{0.92}$ (0.20 μW cm$^{-1}$ K$^{-2}$). W doping and Se vacancies enhance carrier concentration and electrical conductivity in the whole temperature range. It is worth to note that the electrical conductivity abruptly

increases with temperature above 475 K due to the largely increased carrier concentration and carrier mobility. W doping improves Seebeck coefficient through creating resonant energy level. These synergistic effects help to achieve significantly enhanced *PF*. The comparison of the *PF* of SnSe$_{0.92}$ + 0.03WCl$_6$ with those of reported *n*-type polycrystalline SnSe is displayed in Supplementary Fig. 8. Although the *PF* of SnSe$_{0.92}$ + 0.03WCl$_6$ is slightly lower than those of SnSe$_{0.92}$ + 0.01MoCl$_5$ and Sn$_{1.08}$Se−0.13PbTe[42,44], it is significantly higher than those of Sn$_{0.94}$Bi$_{0.06}$Se[39], Sn$_{0.97}$Pb$_{0.03}$Se$_{0.89}$I$_{0.06}$−0.01WSe$_2$[43], SnSe$_{0.95}$−0.04BiCl$_3$[37], and SnSe$_{0.95}$−0.005CeCl$_3$[38]. To better objectively evaluate the intrinsic electrical transport properties of WCl$_6$ doped SnSe$_{0.92}$, the weighted mobility ($\mu_W$) was calculated according to the following formula[59]:

$$\mu_W = \frac{3h^3\sigma}{8\pi e(2m_e k_B T)^{3/2}}\left[\frac{\exp\left[\frac{|S|}{k_B/e}-2\right]}{1+\exp\left[-5\left(\frac{|S|}{k_B/e}-1\right)\right]} + \frac{\frac{3}{\pi^2}\frac{|S|}{k_B/e}}{1+\exp\left[5\frac{|S|}{k_B/e}-1\right]}\right]$$

(4)

**Table 1 | Energy differences among the conduction band maxima and the energy band gaps (E$_g$) in SnSe, SnSe$_{0.92}$, and Sn$_{0.963}$W$_{0.027}$Se$_{0.92}$**

|  | SnSe | SnSe$_{0.92}$ | Sn$_{0.963}$W$_{0.027}$Se$_{0.92}$ |
|---|---|---|---|
| $\Delta E^{CBM1-CBM2}$ (eV) | 0.13 | 0.11 | 0.08 |
| $\Delta E^{CBM1-CBM3}$ (eV) | 0.48 | 0.30 | 0.24 |
| E$_g$ (eV) | 0.61 | 0.45 | 0.60 |

$\Delta E^{CBM1-CBM2}$ and $\Delta E^{CBM1-CBM3}$ indicate the energy differences from the CBM1 (along the Y–X direction) to the second (at the Γ point, CBM2) and the third (along the Z–U direction, CBM3) maxima, respectively.

where $k_B$ is the Boltzmann constant, $h$ is the Planck constant, $m_e$ is the electron mass, and $e$ is the electron charge. The SnSe$_{0.92}$ + 0.03WCl$_6$ sample possesses the highest $\mu_W$ among all the samples (Fig. 3d), identifying its excellent electrical transport properties.

## Thermal transport properties

Figure 5a shows the temperature dependence of total thermal conductivity ($\kappa_T$) of SnSe$_{0.92}$ + x WCl$_6$ (x = 0.01, 0.02, 0.03, 0.04) parallel to the pressing direction. As a result of strong phonon-phonon Umklapp scattering, $\kappa_T$ decreases with increasing temperature. $\kappa_T$ of SnSe$_{0.92}$ + x WCl$_6$ is lower than that of SnSe$_{0.92}$ in the whole temperature range, where SnSe$_{0.92}$ + 0.03WCl$_6$ exhibits the lowest $\kappa_T$ of 0.27 W m$^{-1}$ K$^{-1}$ at 723 K. The electronic thermal conductivity ($\kappa_e$) is estimated according to the Wiedemann-Franz law $\kappa_e = L\sigma T$ (Supplementary Fig. 9a), where $L$ is the Lorenz number calculated from the SPB univalent energy band model (Supplementary Fig. 9b)[1]. The lattice thermal conductivity ($\kappa_L$) is attained by subtracting $\kappa_e$ based on the formula $\kappa_L = \kappa_T - \kappa_e$. $\kappa_L$ of all WCl$_6$ doped samples markedly reduced in the whole temperature range. SnSe$_{0.92}$ + 0.03WCl$_6$ exhibits the lowest $\kappa_L$ among all investigated samples. The lowest $\kappa_L$ as low as 0.24 W m$^{-1}$ K$^{-1}$ was obtained at 773 K in the SnSe$_{0.92}$ + 0.03WCl$_6$ sample (Fig. 5b). As a matter of fact, the lattice thermal conductivities (from 300 to 823 K) of SnSe$_{0.92}$ + 0.03WCl$_6$ are lower than those of other reported *n*-type polycrystalline SnSe systems (Fig. 5c)[37,38,41,43,44]. As shown in Fig. 5b, SnSe$_{0.92}$ + 0.04WCl$_6$ sample shows higher $\kappa_L$ than that of SnSe$_{0.92}$ + 0.03WCl$_6$. Upon doping 3%WCl$_6$ into SnSe$_{0.92}$, we observed the presence of W- and Cl-rich nanoprecipitates (Fig. 6). This indicates that the solubility of WCl$_6$ is lower than 3%. Thus, the introduction of a higher concentration of WCl$_6$ (such as 4%) would result in additional WCl$_6$ precipitates in the matrix. Given that the melting point of WCl$_6$ is

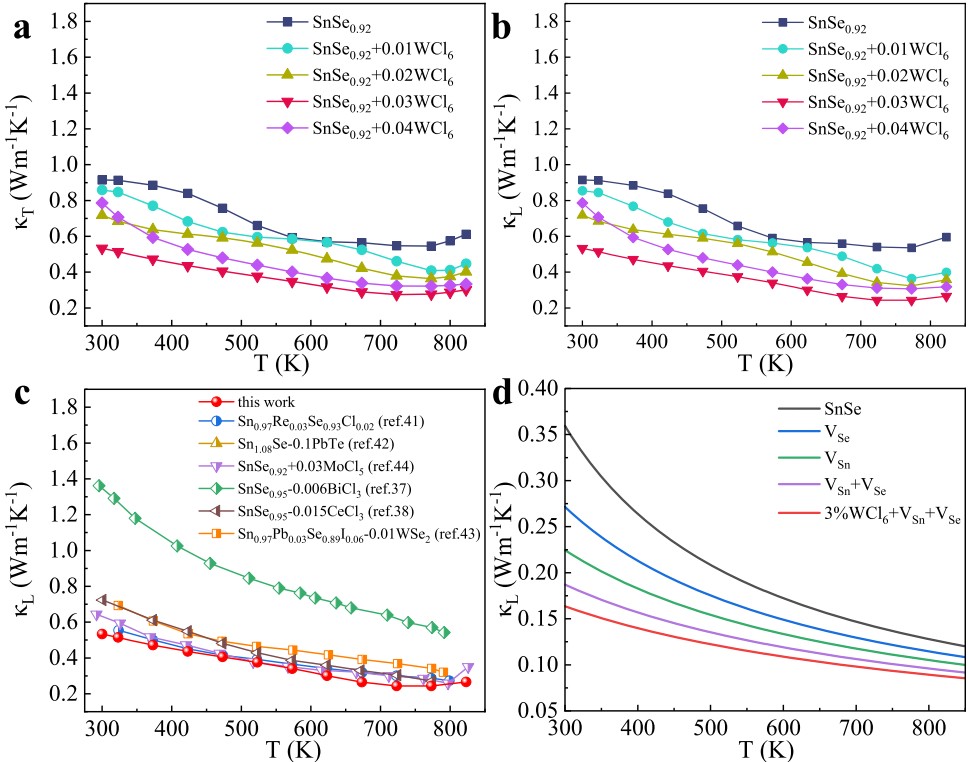

**Fig. 5 | The temperature dependence of thermal transport properties for SnSe$_{0.92}$ + x WCl$_6$. a** Total thermal conductivity ($\kappa_T$), **b** lattice thermal conductivity ($\kappa_L$), **c** comparison of $\kappa_L$, **d** calculated lattice thermal conductivities of SnSe compounds based on the Callaway's model. The black, red, green, orange, and blue lines represent the lattice thermal conductivities of pristine SnSe, V$_{Se}$ (with 8% Se vacancy in SnSe), V$_{Sn}$ (with 14% Sn vacancy in SnSe), V$_{Sn}$ + V$_{Se}$ (with 8% Se and 14% Sn vacancies in SnSe) and 3%WCl$_6$ + V$_{Sn}$ + V$_{Se}$ (with 8% Se and 14% Sn vacancies plus 3% WCl$_6$ doping in SnSe), respectively.

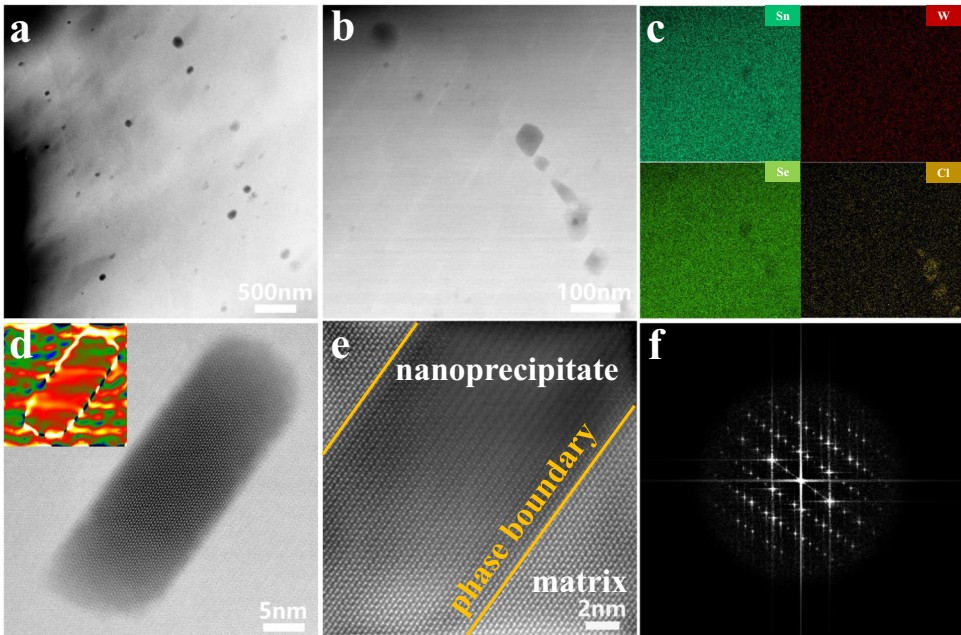

**Fig. 6 | Multiple-scale microstructure of SnSe$_{0.92}$ + 0.03WCl$_6$. a, b** HAADF-STEM images. **c** Energy dispersive spectroscopy (EDS) elemental mapping of (**b**), showing W and Cl enrichment. **d** HADDF-STEM image of a single nanoprecipitate, the inset is the corresponding strain mapping of (**d**). **e** High-magnification HAADF-STEM image focusing on the phase boundary, **f** the corresponding fast Fourier transform (FFT) image.

lower than that of SnSe, the incorporation of WCl$_6$ as a sintering additive can expedite melting and lead to grain coarsening. As shown in Supplementary Fig. 10, the grain sizes of SnSe$_{0.92}$ + 0.04WCl$_6$ are much larger than those of SnSe$_{0.92}$ + 0.03WCl$_6$. Based on the Debye-Callaway model, the $\tau_B^{-1} = \frac{v}{d}$, the larger grain size in the SnSe$_{0.92}$ + 0.04WCl$_6$ matrix corresponds to the larger of the phonon relaxation time $\tau_B$ or the weaker boundary-phonon scattering effect. Therefore, SnSe$_{0.92}$ + 0.04WCl$_6$ sample exhibits the higher lattice thermal conductivity than that of SnSe$_{0.92}$ + 0.03WCl$_6$ due to grain coarsening.

**Microscopic structure and Raman spectral analysis of phonon scattering**

To disclose underlying mechanism of remarkably low $\kappa_L$, we performed microstructural analysis utilizing scanning transmission electron microscopy (STEM) and Raman spectroscopy to elucidate the phonon scattering process. High-angle annular dark-field (STEM-HAADF) images of SnSe$_{0.92}$ + 0.03WCl$_6$ (presented in Fig. 6a, b) depict numerous nanoprecipitates dispersed within the matrix, exhibiting sizes ranging from approximately 10-100 nm. STEM energy dispersive spectroscopy (STEM-EDS) elemental mapping (Fig. 6c) illustrates a higher abundance of Cl and W within the nanoprecipitates, while Sn and Se are more prevalent in the matrix. Quantitative EDS analysis (Supplementary Fig. 11) further supports these findings, indicating a significant increase in the relative intensities of W and Cl within the nanoprecipitates. It is postulated that these nanoprecipitates consist of W- and Cl-rich SnSe phases, exhibiting slightly different chemical compositions compared to the matrix but possessing a similar crystal structure. Individual nanoprecipitates are visualized in the STEM-HAADF image (Fig. 6d), with the corresponding strain field distribution depicted using geometric phase analysis (inset of Fig. 6d), revealing a notable stress-strain at the phase interface. A high-magnification STEM-HAADF image (Fig. 6e) focusing on the phase boundary displays a nearly coherent interface with the matrix, and its corresponding fast Fourier transform (FFT) image showcases the standard orthorhombic SnSe structure (Fig. 6f). The continuous arrangement of the lattice between the matrix and W- and Cl-rich nanoprecipitates, due to their similar chemical composition and crystal structure, which proves that the successful incorporation of endotaxial nanoprecipitates in SnSe$_{0.92}$ + 0.03WCl$_6$. To provide a detailed insight into the atomic structure of the material, a representative atomically-resolved STEM-HAADF image of the matrix is presented in Fig. 7a, allowing for the quantitative analysis of the individual atomic columns. Its atomic coordinates and intensity values are accurately determined, as depicted in the inset of Fig. 7a, where Sn atoms are highlighted in red and Se atoms in green. Weak-intensity sections of the Se atom columns (Fig. 7b, d) suggest the presence of Se vacancies, while significant weak-intensity regions of Sn atoms columns provide solid evidence of a high-density of Sn vacancies (Fig. 7c, e). Notably, the density of Sn vacancies appears to be significantly higher than that of Se vacancies, as qualitatively observed in Fig. 7d, e. The presence of SnO promotes the formation of Sn vacancies in the investigated samples. Endotaxial W- and Cl-rich nanoprecipitates serve as scattering centers for intermediate-wavelength phonons, leading to a reduction in the lattice thermal conductivity while preserving carrier transport properties. In addition, divacancy defects (comprising Se and Sn vacancies) disrupt translation symmetry by causing missing atoms and interatomic linkages, acting as scattering centers for short-wavelength phonons, resulting in substantial reduction of lattice thermal conductivity[60]. As a result, a remarkably low lattice thermal conductivity (0.24 W m$^{-1}$ K$^{-1}$) is achieved in SnSe$_{0.92}$ + 0.03WCl$_6$ sample, benefiting from the combined effect of a high density of endotaxial W- and Cl-rich nanoprecipitates and divacancy defects.

To understand the phonon scattering mechanisms of those defects, we have conducted calculations on the lattice thermal conductivities of SnSe compounds based on the Callaway's model (Fig. 5d)[61,62]. The Gruneisen parameter (2.83), Debye temperature (72 K), and phonon velocity (1936 m/s) were sourced from ref. 63. By introducing Se vacancies and Sn vacancies in the SnSe matrix, their lattice thermal conductivities (at 300 K) exhibit reductions of 22% and 37%, respectively, in comparison to the pristine SnSe compound. Notably, the co-existence of divacancy defects in the matrix leads to a substantial suppression of lattice thermal conductivity by approximately 48% at 300 K relative to that of SnSe alone. The calculated

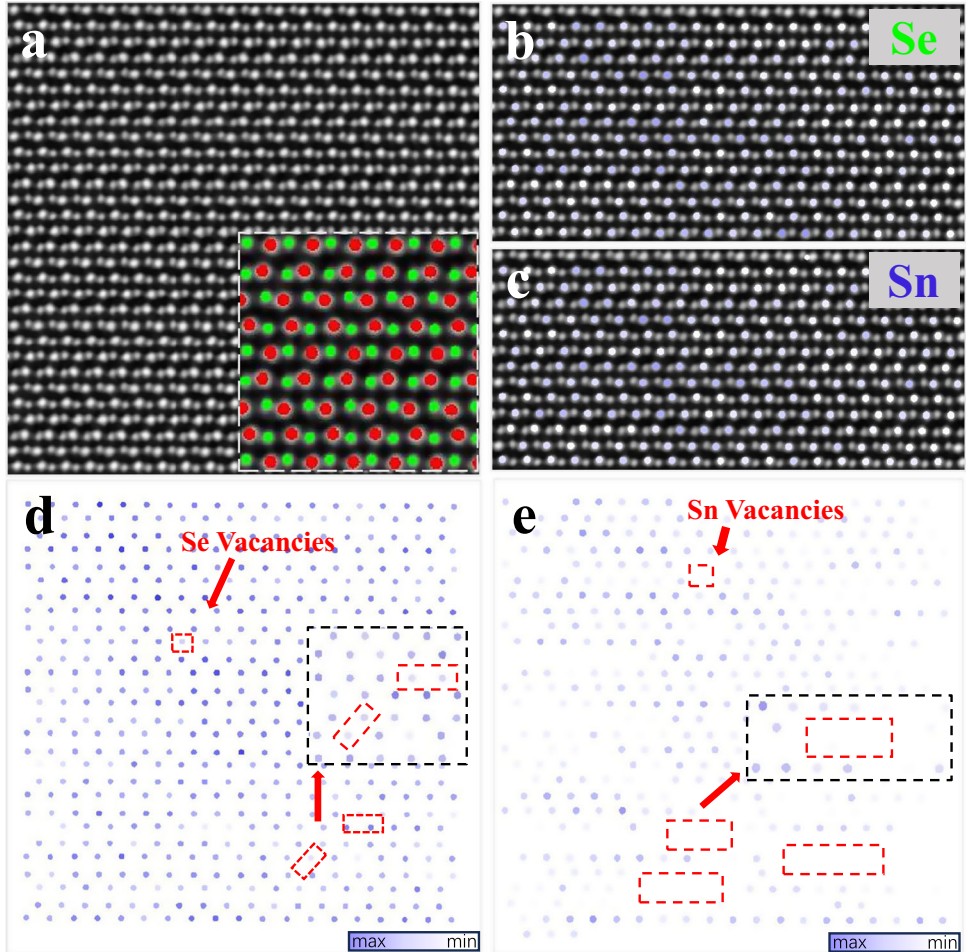

**Fig. 7 | Atomic-scale point defects (vacancies). a** The atomically-resolved STEM HAADF image of the matrix, the inset shows the atomic coordinates of (**a**), in red and green are Sn and Se atoms, respectively, **b, c** the intensity mapping of Se atom and Sn atom columns overlaid on the STEM HAADF image, **d, e** the intensity mapping of Se and Sn, showing Se vacancies and Sn vacancies.

results predict that divacancy defects remarkably reduce the lattice thermal conductivity. Moreover, the incorporation of 3%WCl$_6$ in SnSe further contributes to the depression of the lattice thermal conductivity. Consequently, the combined effects of divacancy and WCl$_6$ doping play pivotal roles in the reduction of the lattice thermal conductivity of SnSe.

The phonon dynamics is essential to investigate as they are bond sensitive and play vital role in the thermal transport properties. SnSe being a member of point group $D_{2h}$ crystallizes in a simple orthorhombic crystal structure with 8 atoms per unit cell. The crystal structure of SnSe allows total of 24 phonon modes, among them 3 are acoustic phonon modes and 21 are optical phonon modes[64]. To understand the effect of WCl$_6$ doping on phonon lifetime in polycrystalline SnSe, room temperature Raman scattering experiments were performed with excitation laser energy of 785 nm. The room temperature Raman spectra of SnSe as a function of doping content are comparably plotted in Fig. 8a. In backscattering configuration, the Raman spectra of polycrystalline SnSe clearly resolved $B_{3g}$, $A_g^2$, and $A_g^3$ phonon modes at 108, 128, and 149 cm$^{-1}$, respectively[65]. Besides these well-reported phonon modes, an additive combination $A_g^0 + A_g^3$ phonon mode was clearly observed at 182 cm$^{-1}$ whose frequency is the sum of $A_g^0$ (33 cm$^{-1}$) and $A_g^3$ (150 cm$^{-1}$) phonons. The origin of this additive mode can be attributed to the polycrystalline nature of SnSe and backscattering configuration of the Raman scattering experiment. It is obvious from the Fig. 8a that with the increase in dopant content, the Raman intensity of all these phonons suppressed and the line widths of

these phonons broadened. The Raman line width (Γ) is associated with lifetime (τ) of the phonon by the relation $\frac{1}{\tau} = \frac{\Gamma}{\hbar}$, where Γ represents the FWHM of a peak in unit of cm$^{-1}$, ℏ is the Plank constant and its value is $5.3 \times 10^{-12}$ cm$^{-1}$ s[66]. The intensity and life time variation of the strongest phonon $A_g^3$ are shown in Fig. 8b. In comparison of pure SnSe the intensity of $A_g^3$ decreased about 50% for SnSe+0.04WCl$_6$ and its life time decreased from 0.73 to 0.55 Ps. To further insight the doping effect of all observed phonon, the analysis of relative intensity and relative life time ratios $A_g^3/A_g^0 + A_g^3$ and $A_g^3/B_{3g}$ was carried out and displayed in Fig. 8c, d. The decreasing phonon intensity and lifetime ratios are indicative of optical phonon softening with increased WCl$_6$ doping content. In Fig. 8c, d, the decreasing intensity and lifetime ratios $A_g^3/A_g^0 + A_g^3$ and $A_g^3/B_{3g}$ signify the rapid softening of $A_g^3$ phonon in comparison to other $B_{3g}$ and $A_g^0 + A_g^3$ phonon modes. The Raman scattering experiments confirmed that the WCl$_6$ doping leads to optical phonon softening and strengthens anharmonicity, facilitating to obtain ultralow lattice thermal conductivity in SnSe$_{0.92}$ + 0.03WCl$_6$.

### Electro-phonon synergies and thermoelectric properties

Figure 9a shows the ratio of $\mu_W$ to $\kappa_L$ ($\mu_W/\kappa_L$), the values of SnSe$_{0.92}$ + x WCl$_6$ are higher than those of SnSe$_{0.92}$ in the whole temperature range, indicating that WCl$_6$ doping is beneficial in promoting the synergistic optimization of electron and phonon transport. In particular, SnSe$_{0.92}$ + 0.03WCl$_6$ shows the highest value of $\mu_W/\kappa_L$. For the more objective evaluation of the synergistic modulation of the electrical and thermal transport

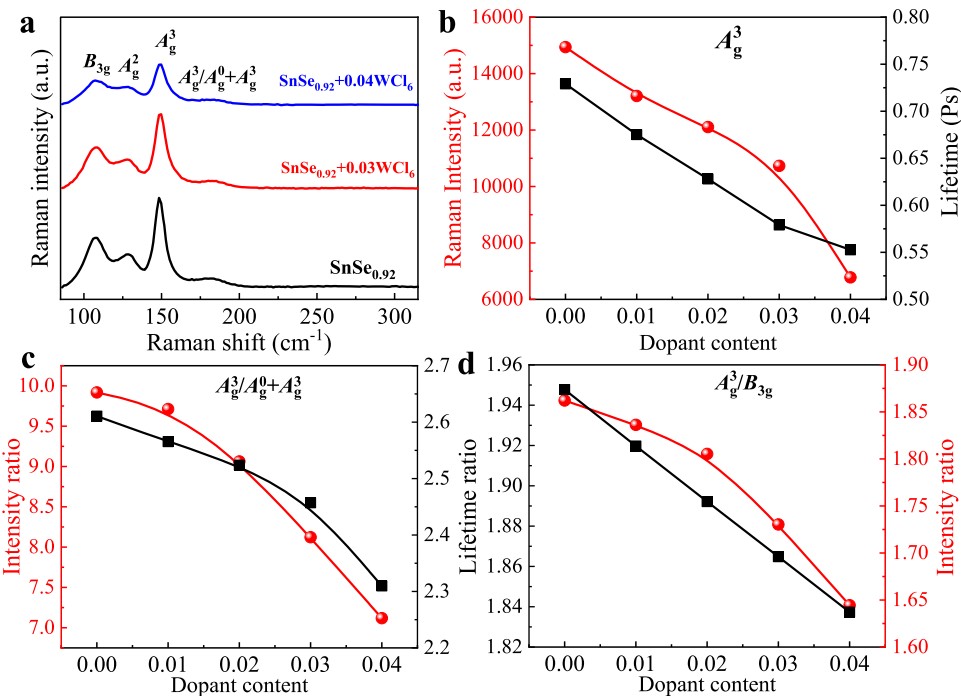

**Fig. 8 | Room temperature recorded Raman spectra of polycrystalline SnSe. a** Raman spectra as a function of dopant content, **b** Absolute intensity and lifetime of $A_g^3$, **c** Relative intensity and life time ratio $A_g^3/A_g^0 + A_g^3$, **d** Relative intensity and lifetime ratio $A_g^3/B_{3g}$.

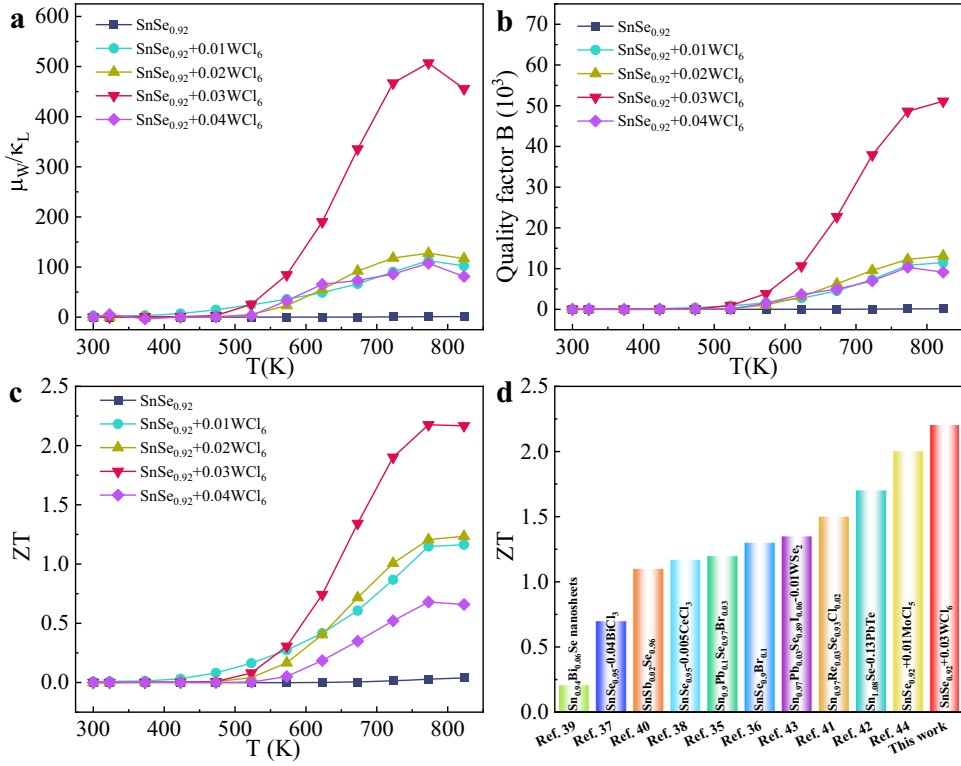

**Fig. 9 | Electro-phonon synergies and dimensionless figure-of-merit ZT as a function of temperature for *n*-type SnSe$_{0.92}$ + x WCl$_6$. a** $\mu_W/\kappa_L$, **b** quality factor (*B*), **c** *ZT* along the pressing direction. **d** Comparisons of *ZT* for SnSe$_{0.92}$ + 0.03WCl$_6$ with *n*-type polycrystalline SnSe-based systems.

properties of these samples, the dimensionless quality factor (*B*) was calculated according to the following formula (Fig. 9b)[67]:

$$B = 9\frac{\mu_W}{\kappa_L}\left(\frac{T}{300}\right)^{5/2} \qquad (5)$$

*B* is significantly enhanced in SnSe$_{0.92}$ + x WCl$_6$ for all samples, SnSe$_{0.92}$ + 0.03WCl$_6$ exhibits an excellent *B* factor over a wide temperature range, predicting that SnSe$_{0.92}$ + 0.03WCl$_6$ will have a high *ZT* value. As shown in Fig. 9c, all the WCl$_6$ doped SnSe$_{0.92}$ show noticeable enhancement of *ZT* compared to SnSe$_{0.92}$. The highest peak *ZT* of 2.2 at 773 K is obtained in SnSe$_{0.92}$ + 0.03WCl$_6$, which benefits from the

combination of significantly enhanced *PF* and suppressed lattice thermal conductivity. The reproducible measurements of Supplementary Fig. 12 demonstrate the good experimental reproducibility of this work. To the best of our knowledge, this is a record-high value among utmost reported *n*-type polycrystalline SnSe (Fig. 9d)[35–44]. It is worth to note that the reported high performance outperforms other well-known *n*-type thermoelectric systems as well[45–51].

In conclusion, this work demonstrated a promising strategy to significantly improve thermoelectric performance of *n*-type polycrystalline SnSe. Aberration-corrected STEM observations have revealed the presence of divacancy defects (Se vacancies and Sn vacancies) within the matrix. These divacancy defects disrupt translation symmetry by causing the absence of atoms and interatomic linkages, leading to reduction of lattice thermal conductivity. The combination of divacancy defects and endotaxial W- and Cl-rich nanoprecipitates creates a multiscale microstructure that effectively scatters phonons, leading to an ultralow lattice thermal conductivity. The Raman scattering experiments have confirmed that the $WCl_6$ doping leads to optical phonon softening and strengthens anharmonicity. Furthermore, it has been shown that W doping introduces resonance levels within the conduction bands and increase DOS in the electronic structure of SnSe, consequently boosting the Seebeck coefficient. More importantly, the optimization of carrier concentration through $WCl_6$ doping and Se vacancies, ensures superior electrical conductivity. The highest *PF* of 7.95 μW cm$^{-1}$ K$^{-2}$ is achieved in $SnSe_{0.92} + 0.03WCl_6$ at 773 K. As a result, a record-breaking peak *ZT* of 2.2 at 773 K has been realized in *n*-type $SnSe_{0.92} + 0.03WCl_6$ polycrystals by decoupling electron and phonon transport.

## Methods

### Sample fabrication
A series of polycrystalline $SnSe_{0.92} + x$ mol $WCl_6$ (x = 0.01, 0.02, 0.03, 0.04) samples were synthesized. Sn powder (99.99%, Aladdin), Se powder (99.99%, Aladdin), and $WCl_6$ powder (99.99%, Aladdin) were vacuum-sealed in a quartz tube according to the above stoichiometric ratio. The sealed tube was then heated to 1223 K within 20 h, and held at this temperature for 4 h, followed by an 8-hour cooling process to bring it down to 873 K. Finally, the sample was allowed to cool to room temperature over a period of 15 h, after which the obtained ingot was ground into powder. In the final step, the powder was pressed into dense cylindrical samples with a diameter of 10 mm using spark plasma sintering (SPS) (HPD 10, FCT System GmbH) at 723 K and 50 MPa pressure for 7 mins.

### Characterization
An X-ray diffraction (XRD) instrument (Bruker D8 Advance) with Cu K*α* radiation was used to analyze the phase structure of $SnSe_{0.92} + x$ mol $WCl_6$ (x = 0.01, 0.02, 0.03, 0.04) samples. An X-ray photoelectron spectroscopy (XPS) instrument (Krayos AXIS Ultra DLD) was used to analysis the valence of element W and Sn. Measurements of the pair distribution function (PDF) using synchrotron X-ray were carried out at the BL13SSW beamline within the Shanghai Synchrotron Radiation Facility (SSRF). The X-ray radiation had a center energy of 50.00 keV, corresponding to a wavelength of $\lambda = 0.2480$ Å. Scattered X-rays were collected using a flat area detector (Mercu1717V). Calibration and integration of 2D signals were performed using DIOPTAS. Corrections for environmental scattering, incoherent and multiple scattering, polarization, and absorption were implemented through PDFGETX2, with $Q_{max}$ set at 20.7 Å$^{-1}$. Aberration-corrected scanning transmission electron microscopy (STEM) and energy-dispersive X-ray spectroscopy (EDS) investigation were conducted on JEM-ARM300F2 and JEM-NEOARM equipped with cold field-emission guns and spherical aberration correctors, at Instrumental Analysis Center of Xi'an Jiaotong University. The phonon scattering in polycrystalline SnSe was studied using the Raman scattering experiment. Renishaw inVia Raman microscope was used in backscattering configuration to record the Raman spectra. The samples were excited by laser line of 785 nm wavelengths to measure resonance Raman spectra. The laser power was set to 0.5 mW to avoid serious local heating effect. The backscattered light was collected by a 50× objective lens dispersed by 1200 lines/mm for 785 nm laser.

### Density-functional theory calculations
The density-functional theory[68] with the projector augmented wave method[69–71] based Vienna ab initio simulation package (VASP) was used for the electronic band structure calculations. The exchange and correlation functional was approximated using the generalized gradient approximation of Perdew−Burke−Ernzerhof (GGA-PBE)[68]. The energy cutoff of 400 eV was used to truncate the expansion of plane-wave, and the Monkhorst-Pack[72] meshes with a roughly constant *k*-points density of 30 Å$^3$ were used to sample the Brillouin zones. The threshold to converge the total energy was set as 10$^{-5}$ eV. To fully optimize the geometry structures, the components of forces and the stress tensor were below 0.01 eV/Å$^2$ and 0.2 kbar, respectively. To understand the effects of defects (the W doping and Se vacancy) on the electronic structures, we simulated the experimentally suggested the concentrations of W doping (~3%) and Se vacancy (~8%) in SnSe: building a large SnSe (3 × 3 × 1) supercell (*a* = 12.65 Å, *b* = 13.58 Å, *c* = 11.81 Å, containing 36 Sn and 36 Se atoms, Supplementary Fig. 6b), randomly taking three Se out the cell ($SnSe_{0.92}$) and replacing one Sn atom by W ($Sn_{0.963}W_{0.027}Se_{0.92}$). The most stable defect structures used for further electronic structure studies (Fig. 4d and Supplementary Fig. 4) were achieved by selecting among all possible defect configurations with the lowest energetics. The band structures of a supercell generated based on the corresponding primitive cell undergone the well-known band folding. To clearly investigate defect effects on the band structure, the band unfolding methodology (the BandUP code) was used to recover the effective primitive picture along the high symmetry directions in the primitive cell Brillouin zone[73].

### Thermoelectric property measurement
Both the thermal and electrical transport properties of all the samples were measured parallel to the pressing direction. The electrical conductivity ($\sigma$) and Seebeck coefficient (*S*) from 300 to 823 K were measured on an Ulvac-Riko ZEM-3 instrument system under a helium atmosphere. A Netzsch LFA-457 instrument was used to measure the thermal diffusivity coefficient (*D*) under an argon atmosphere. The density ($\rho$) (Supplementary Table 4) was measured on a density meter (ME204E), and the specific heat ($C_p$) was taken from the ref. 22, then the thermal conductivity was calculated from the formula $\kappa_T = DC_p\rho$. Carrier concentration (*n*) and mobility ($\mu$) at room temperature were measured on a Hall measurement instrument (HMS-3000). The temperature-dependent Hall coefficient ($R_H$) was measured by the van der Pauw method on a Hall-effect measurement system (HMS8400, Lake Shore Cryotronics). The Hall carrier concentration (*n*) and carrier mobility ($\mu$) were calculated using $n = 1/eR_H$ and $R_H/\rho$, respectively. For both $\sigma$ and *S*, the uncertainty is ~5%. The uncertainty is ~12%, where *D* is ~5%, $C_p$ is ~5%, and $\rho$ is ~2%. In total, *ZT* has a combined uncertainty of ~20%.

### Reporting summary
Further information on research design is available in the Nature Portfolio Reporting Summary linked to this article.

## Data availability
The authors declare that all data supporting the findings of this study are available within the article and its Supplementary Information files or from the corresponding author.

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

## Acknowledgements

The work was supported by the National Natural Science Foundation of China (No. 52071182 to G.D.T. and 52172128 to H.J.W.), "Qinglan Pro-ject" of the Young and Middle-aged Academic Leader of Jiangsu Pro-vince (to G.D.T.), the Fundamental Research Funds for the Central Universities (No. 30921011107 to G.D.T), the Postgraduate Research & Practice Innovation Program of Jiangsu Province (KYCX23_0457 to Y.R.G.), the National Key R&D Program of China (2021YFB3201100 to H.J.W.), and the Distinguished Expert of Taishan Scholar (No. tstp20221124 to Y.S.Z.). We thank Mrs. Yang Zhang from Instrument Analysis Center of Xi'an Jiaotong University for the assistance with electron microscopy characterization. We also thank the staff of beamline BL13SSW at Shanghai Synchrotron Radiation Facility for experiments supports.

## Author contributions

Y.R.G. and W.D. prepared samples, analyzed data, and wrote the paper. B.C.L. and H.J.W. accomplished the microstructural characterizations. X.M.Z. and Y.S.Z. carried out the DFT calculations. H.Z. helped the syn-chrotron pair distribution function analysis. P.Y., Q.T.Z., Y.Q.L., X.Q.H., S.H.Z., and D.L. helped measure the properties. Y.N.L. helped prepare samples. M.F.I. helped Raman spectra analysis. G.D.T. initiated the idea, conceived and designed the experiments, revised the manuscript. All authors discussed the results.

## Competing interests

The authors declare no competing interests.
