## [Peer Review File · Nature Communications]

Divacancy and Resonance Level Enables High Thermoelectric Performance in n-Type SnSe PolycrystalsReviewers' comments:

Reviewer #1 (Remarks to the Author):

The paper presents significantly high n-type ZT values for SnSe; however, the authors' explanation of the results is lacking. This inadequacy means that the paper falls short of meeting the standard expected by Nature Communications. Consequently, the review cannot recommend this paper for further consideration. The specific comments are outlined below:

1. The authors assert that the high power factor is due to a large Seebeck coefficient. However, based on Figure 3(a)(b)(c), it appears that the change in power factor aligns more closely with electrical conductivity rather than the Seebeck coefficient. Although the Seebeck coefficient for SnSe_{0.92}+0.03WCl₆ is notably higher than that of SnSe_{0.92}, their power factors at low temperatures are similar. As temperature increases, the power factor begins to rise due to increased electrical conductivity, while the Seebeck coefficient shows minimal change. This discrepancy leads the reviewer to disagree with the conclusion.

2. Given the flawed conclusion, the DFT analysis in Figure 4 becomes inconsequential. In actuality, the Density of States (DOS) for both SnSe_{0.92} and SnSe_{0.963}W_{0.027}Se_{0.92} exhibits a sharp increase around the Fermi level. Even with the current findings, the rise in DOS above the Conduction Band Minimum (CBM) is not evident, whereas the increase in the Seebeck coefficient is clearly pronounced. Therefore, the explanation for the Seebeck coefficient appears unfounded.

Reviewer #2 (Remarks to the Author):

This paper reports a record-high peak ZT of 2.2 at 773 K in n-type SnSe by WCl₆ doping and Se vacancies. This is a systematic study. However, for the publication in Nature communications, it seems that more mechanism and experimentation reinforcement is still needed, and there is a lack of explanation of the experiments and mechanisms.

1. In Table S1, the Rietveld refinement result does not show any insufficient occupancy in Sn/W site or Se/Cl site. Whether the existence of divacancy defects can be proved by the Rietveld refinement from the XRD data?
2. In Figure 2f, the XPS peaks with binding energy of ca. 493.7 and 485.3 eV are attributed to SnO. How does SnO affect the thermoelectric properties of n-type SnSe?
3. In Table 1, the carrier concentration increases to $4.174 \times 10^{18} \text{ cm}^{-3}$ after 1% WCl₆ doping, but it decreases sharply for the samples of x0.02, further discussion should be given.
4. The sample of "SnSe_{0.92} + 0.03WCl₆" has a similar carrier concentration to the SnSe_{0.92}. Thus, the enhanced of "SnSe_{0.92} + 0.03WCl₆" can be mainly attributed to enhanced mobility, the reasons for this need to be discussed.
5. In order to clarify the effect of different phonon scatterings on κ_L , please calculate the theoretical lattice thermal conductivity based on Callaway's model.
6. The test direction for the sintered samples should be presented for the transport properties measurement since SnSe has a layered like crystal structures.
7. The English writing needs to be more concise for better readability.

Reviewer #3 (Remarks to the Author):

The authors systematically present WCl₆ doped SnSe to enhance the carrier concentration and obtain a large Seebeck coefficient. Microstructural characterization is also carried out to elucidate the reason for the low lattice thermal conductivity. They enabled polycrystalline SnSe to achieve a larger ZT value of 2.2 at 773 K. The results are summarized in the following table. The experiments are informative and fulfilling but suffer from the following problems.

1 There is at least one terminological error in the manuscript, for example, in the abstract, "ultralow lattice conductivity" should be "ultralow lattice thermal conductivity". Please check the manuscript carefully.

2 No explanation is given as to why the Seebeck coefficient of SnSe_{0.92}+0.04WCl₆ decreases with increasing temperature while all other trends increase with increasing temperature in Figure 3B.

3 The trends of total thermal conductivity and lattice thermal conductivity in Fig. 5 are similar due to the low electronic thermal conductivity, and it is sufficient to use the same comparison chart when making the comparative illustration without repeating the explanation.

February 17 2024

Many thanks for your correspondence concerning the Reviewers' comments on our manuscript (NCOMMS-24-02607-T) entitled "Divacancy and Resonance Level Enables High Thermoelectric Performance in n-Type SnSe Polycrystals". We are grateful for the Reviewers' positive remarks and constructive suggestions for improving our manuscript. We would also like to express our appreciation for your valuable input. We have modified our manuscript accordingly and the point-by-point response to the Reviewers' comments is given below.

Reviewer: 1

Comment: 1. The authors assert that the high power factor is due to a large Seebeck coefficient. However, based on Figure 3(a)(b)(c), it appears that the change in power factor aligns more closely with electrical conductivity rather than the Seebeck coefficient. Although the Seebeck coefficient for SnSe_{0.92}+0.03WCl₆ is notably higher than that of SnSe_{0.92}, their power factors at low temperatures are similar. As temperature increases, the power factor begins to rise due to increased electrical conductivity, while the Seebeck coefficient shows minimal change. This discrepancy leads the reviewer to disagree with the conclusion.

Response:

Thanks for your comment. The power factor (*PF*) can be defined as $S^2\sigma$. Thus, it is necessary to consider the effect of S^2 (NOT S) and σ on *PF*. Taking the SnSe_{0.92}-3% WCl₆ sample as an example, we plot σ vs T and S^2 vs T, as shown in Fig. R1. We can find that S^2 and σ show similar change trend with *PF* (Fig. R1). As temperature increases, the power factor begins to rise due to increased electrical conductivity and Seebeck coefficient. This indicates that the high power factor results from simultaneously enhanced Seebeck coefficient and electrical conductivity. In general, S and σ are coupled by the carrier concentration, and can be expressed as¹

$$\sigma = en\mu$$

S is the Seebeck coefficient, k_B is the Boltzmann constant, h is the Planck constant, e is the electron charge, n is the carrier concentration, and m^* is the effective mass and μ is the carrier mobility. This demonstrates the inverse correlation between carrier concentration and Seebeck coefficient. The optimization of carrier concentration to increase σ usually leads to the significantly reduced Seebeck coefficient, which greatly

limits the ability for further optimizing electrical transport properties and power factor. Such coupling represents one major challenging in the field of thermoelectrics. However, we can improve the Seebeck coefficient and power factor by introducing the resonance level and enhancement of DOS created by W doping even though the carrier concentration increases. As compared with pristine sample, $\text{SnSe}_{0.92}+0.03\text{WCl}_6$ sample exhibit significantly enhanced Seebeck coefficient, which is very important for obtaining high thermoelectric performance in our samples. This conclusion is one of the novelties in our manuscript.

Fig. R1 The temperature dependence of (A) σ , (B) S^2 for $\text{SnSe}_{0.92}+0.03\text{WCl}_6$ sample.

Comment: 2. Given the flawed conclusion, the DFT analysis in Figure 4 becomes inconsequential. In actuality, the Density of States (DOS) for both $\text{SnSe}_{0.92}$ and $\text{SnSe}_{0.963}\text{W}_{0.027}\text{Se}_{0.92}$ exhibits a sharp increase around the Fermi level. Even with the current findings, the rise in DOS above the Conduction Band Minimum (CBM) is not evident, whereas the increase in the Seebeck coefficient is clearly pronounced. Therefore, the explanation for the Seebeck coefficient appears unfounded.

Response:

The density of states (DOS) calculated using Density Functional Theory (DFT) is crucial for comprehending the underlying physical mechanisms. Although the DOS of $\text{SnSe}_{0.92}$ and $\text{SnSe}_{0.963}\text{W}_{0.027}\text{Se}_{0.92}$ both exhibit distinct peaks near the Fermi level, their respective impacts on the contribution to the Seebeck coefficient are different. From Figs. 4C and 4F in the manuscript (or Fig. R2 below), the distinction in DOS around the Fermi level is evidently apparent. In the case of $\text{SnSe}_{0.92}$, the minor DOS peak lies slightly below the Fermi energy level, originating from the Sn-p energy level. It is evident that the peak is far away from the conduction band minimum (CBM), representing a single defect energy level within the band gap, thus not playing an important role to the *n*-type electron transport. However, in the case of

$\text{SnSe}_{0.963}\text{W}_{0.027}\text{Se}_{0.92}$, a pronounced DOS peak emerges at the Fermi level. Unlike the defect energy level in $\text{SnSe}_{0.92}$, this DOS peak originates from the d energy level of W dopant, forming the resonant energy level with the SnSe matrix around the CBM. Such W-induced DOS peak will deeply involve in the *n*-type electron transport and the increased DOS will significantly boost the Seebeck coefficient. To facilitate a comprehensive comparison of the two DOS profiles, we additionally put them in one plot (Fig. R3). It is apparent that the defect DOS peak in $\text{SnSe}_{0.92}$ is considerably distanced from the CBM in contrast to that of $\text{SnSe}_{0.963}\text{W}_{0.027}\text{Se}_{0.92}$. This discrepancy elucidates the enhancement in Seebeck effect as indicated by experimental observations.

To clarify this point, we have added the followings in the revised manuscript:

“However, since it is below the Fermi level and far away from the conduction band minimum, it is a defect energy level in the band gap and would not contribute to the *n*-type electron transport.” On page 10.

“Unlike the defect energy level in the DOS of $\text{SnSe}_{0.92}$, this DOS peak is from the d energy level of W dopant (Fig. S6C), and it forms the resonant energy state with the SnSe matrix just below the CBM. This W induced DOS peak will involve in the *n*-type electron transport.” On page 11.

Fig. R2 Density of states of $\text{SnSe}_{0.92}$ and $\text{Sn}_{0.963}\text{W}_{0.027}\text{Se}_{0.92}$. The dashed line represents the Fermi energy level.

Fig. R3 Density of states of $\text{SnSe}_{0.92}$ and $\text{Sn}_{0.963}\text{W}_{0.027}\text{Se}_{0.92}$.

Reviewer: 2

Comment: This paper reports a record-high peak ZT of 2.2 at 773 K in n-type SnSe by WCl_6 doping and Se vacancies. This is a systematic study. However, for the publication in Nature communications, it seems that more mechanism and experimentation reinforcement is still needed, and there is a lack of explanation of the experiments and mechanisms.

Response:

Thank you for carefully reviewing and offering insightful suggestions for the enhancement of our manuscript, and we provide a point-to-point response as follows, in the hopes of addressing any uncertainties you may have.

Comment: 1. In Table S1, the Rietveld refinement result does not show any insufficient occupancy in Sn/W site or Se/Cl site. Whether the existence of divacancy defects can be proved by the Rietveld refinement from the XRD data?

Response:

Many thanks for your suggestions. In the initial manuscript, default constraints were employed during the XRD Rietveld refinement process. Specifically, the atomic occupancy stoichiometric ratios for the Wyckoff positions of Sn and Se in SnSe were set to 1. However, these default constraints implied the absence of vacancies in either Sn and Se positions, conflicting with the STEM analysis. During the revision process, we removed this constraint, leading to more precise results of the refinement treatment, as shown in Table R1 and Fig. R4. It was observed that W and Cl replaced Sn and Se, occupying similar Wyckoff positions. The results also indicated the existence of

divacancy defects, with a notably higher vacancy concentration in the Sn positions, which is consistent with our aberration-corrected STEM quantitative analysis.

Revision: “W and Cl substitute Sn and Se and occupy similar Wyckoff positions, respectively, and a certain content of vacancies exists at both Sn and Se positions.” On page 6.

Table S1 and Fig. 2B was revised accordingly.

Table R1. Rietveld refinement details of $\text{SnSe}_{0.92} + 0.03 \text{WCl}_6$.

Atom	Site	x	y	z	Occupancy	FWHM(111)
Sn	4c	0.1187	0.2500	0.0988	0.8451	
W	4c	0.1187	0.2500	0.1124	0.0157	
Se	4c	0.8520	0.2500	0.4867	0.7665	0.2368
Cl	4c	0.8710	0.2500	0.4517	0.1246	

Fig. R4 Rietveld refinement details of the XRD pattern of $\text{SnSe}_{0.92} + 0.03 \text{WCl}_6$.

Comment: 2. In Figure 2f, the XPS peaks with binding energy of ca. 493.7 and 485.3 eV are attributed to SnO. How does SnO affect the thermoelectric properties of n-type SnSe?

Response:

Thank you for your comment. Generally, SnO has a high lattice thermal conductivity than SnSe,² affecting thermal transport properties adversely.² However, in this work, the presence of SnO promotes the formation of Sn vacancies in our samples. Sn

vacancies and Se vacancies (divacancy defect) strongly scattering phonons, leading to significantly reduced lattice thermal conductivity. This is also evidenced by theoretical lattice thermal conductivity based on Callaway's model (Please see our Response to your Comment 5). The related discussion is incorporated into the revised manuscript.

Revision: "The presence of SnO promotes the formation of Sn vacancies in the investigated samples." On page 14.

"Additionally, divacancy defects (comprising Se and Sn vacancies) disrupt translation symmetry by causing missing atoms and interatomic linkages, acting as supplementary scattering centers for short-wavelength phonons, further decreasing lattice thermal conductivity.⁶⁰" On page 14.

Comment: 3. In Table 1, the carrier concentration increases to $4.174 \times 10^{18} \text{ cm}^{-3}$ after 1% WCl₆ doping, but it decreases sharply for the samples of x0.02, further discussion should be given.

Response:

Thanks for your suggestions. A possible mechanism for carrier variation of SnSe_{0.92+0.01WCl₆} and SnSe_{0.92+0.02WCl₆} is given as follows:

The Cl will progressively occupy the Se vacancy with increasing WCl₆ doping content. The introduction of Cl dopants results in an imbalance, the density of Sn vacancies is larger than that of Se vacancies, which is evidenced by aberration-corrected STEM analysis (Fig. R5). The density of Sn vacancies appears to be significantly higher than that of Se vacancies, as observed in Figs. 7D and 7E. This discrepancy is anticipated to culminate in a decline in carrier concentration.

Fig. R5 Atomic-scale point defects (vacancies): (A) the atomically-resolved STEM HAADF image of the matrix, the inset shows the atomic coordinates of (A), in red and green are Sn and Se atoms, respectively, (B, C) the intensity mapping of Se atom and Sn atom columns overlaid on the STEM HAADF image, (D, E) the intensity mapping of Se and Sn, showing Se vacancies and Sn vacancies.

Revision: “The carrier concentration decreases sharply from $x=0.02$. The Cl will progressively occupy the Se vacancies with increasing WCl_6 doping content. The introduction of Cl dopants results in an imbalance, the density of Sn vacancies is larger than that of Se vacancies, which is evidenced by aberration-corrected STEM analysis. This discrepancy is anticipated to culminate in a decline in carrier concentration.” On page 8.

Comment: 4. The sample of “ $SnSe_{0.92} + 0.03WCl_6$ ” has a similar carrier concentration to the $SnSe_{0.92}$. Thus, the enhanced of “ $SnSe_{0.92} + 0.03WCl_6$ ” can be mainly attributed to enhanced mobility, the reasons for this need to be discussed.

Response:

Thanks for your suggestions. In principle, with WCl_6 doping, the carrier mobility of $SnSe_{0.92} + 0.03WCl_6$ is lower than that of $SnSe_{0.92}$ due to the introduced lattice defects and

nanoprecipitates. Thus, the increased electrical conductivity in the WCl_6 doping is ascribed to the enhanced carrier concentrations. From Fig. R6, the pristine $\text{SnSe}_{0.92}$ has the lowest carrier concentration and the low electronic conductivity. Once doping 1% WCl_6 in $\text{SnSe}_{0.92}$, the carrier concentration is strongly increased, and the sample shows the highest electronic conductivity. However, with increasing the WCl_6 content, the carrier concentration is gradually decreased. The reason is given (Response to your comment 3#). If we look closely at the carrier concentrations and electronic conductivities of these samples, we notice that the carrier concentration of $\text{SnSe}_{0.92} + 0.03\text{WCl}_6$ is still higher than that of $\text{SnSe}_{0.92}$, so does the electrical conductivity. It is the $\text{SnSe}_{0.92} + 0.04\text{WCl}_6$ sample has the similar carrier concentration as that of $\text{SnSe}_{0.92}$, and the electrical conductivity is also highly similar. Thus, the trend of these carrier concentrations is in reasonable consistent with the electrical conductivity behavior.

Fig.R6 The carrier concentration and electrical conductivity (σ) of $\text{SnSe}_{0.92} + x\text{WCl}_6$ samples.

Comment: 5. In order to clarify the effect of different phonon scatterings on κ_L , please calculate the theoretical lattice thermal conductivity based on Callaway's model.

Response:

We express our appreciation for the valuable suggestion provided. Following your suggestion, we have conducted calculations on the lattice thermal conductivities of SnSe compounds based on the Callaway's model (Fig. R7). The Gruneisen parameter (2.83), Debye temperature (72 K) and phonon velocity (1936 m/s) were sourced from the literature.³ In addition to the 8% Se vacancy, the XRD refinement has indicated the Sn vacancy concentration as 14% in the $\text{SnSe}_{0.92} + 0.03 \text{WCl}_6$ sample, as shown in Table R1. By introducing Se vacancies and Sn vacancies in the SnSe matrix, their lattice thermal conductivities (at 300 K) exhibit reductions of 22% and 37%, respectively, in

comparison to the pristine SnSe compound. Notably, the coexistence of divacancy defects in the matrix leads to a substantial suppression of lattice thermal conductivity by approximately 48% at 300 K relative to that of SnSe alone. The calculated results predict that divacancy defects remarkably reduce the lattice thermal conductivity. Moreover, the incorporation of 3% WCl_6 in SnSe further contributes to the depression of the lattice thermal conductivity. Consequently, the combined effects of divacancy and WCl_6 doping play pivotal roles in the reduction of the lattice thermal conductivity of SnSe.

We have added the following in the revised manuscript:

“To understand the phonon scattering mechanisms of those defects, we have conducted calculations on the lattice thermal conductivities of SnSe compounds based on the Callaway’s model (Fig. 5D).^{61, 62} The Gruneisen parameter (2.83), Debye temperature (72 K) and phonon velocity (1936 m/s) were sourced from Ref. 63. By introducing Se vacancies and Sn vacancies in the SnSe matrix, their lattice thermal conductivities (at 300 K) exhibit reductions of 22% and 37%, respectively, in comparison to the pristine SnSe compound. Notably, the coexistence of divacancy defects in the matrix leads to a substantial suppression of lattice thermal conductivity by approximately 48% at 300 K relative to that of SnSe alone. The calculated results predict that divacancy defects remarkably reduce the lattice thermal conductivity. Moreover, the incorporation of 3% WCl_6 in SnSe further contributes to the depression of the lattice thermal conductivity. Consequently, the combined effects of divacancy and WCl_6 doping play pivotal roles in the reduction of the lattice thermal conductivity of SnSe.” On page 15.

Fig. R7 Calculated lattice thermal conductivities of SnSe compounds based on the Callaway’s model. The black, red, green, orange and blue lines represent the lattice thermal conductivities of pristine SnSe, V_{Se} (with 8% Se vacancy in SnSe), V_{Sn} (with 14% Sn vacancy in SnSe), $V_{\text{Sn}}+V_{\text{Se}}$ (with 8% Se and 14% Sn vacancies in SnSe) and

3% $\text{WCl}_6 + \text{V}_{\text{Sn}} + \text{V}_{\text{Se}}$ (with 8% Se and 14% Sn vacancies plus 3% WCl_6 doping in SnSe), respectively.

Comment: 6. The test direction for the sintered samples should be presented for the transport properties measurement since SnSe has a layered like crystal structures.

Response:

Thank you for your valuable suggestions. Polycrystalline SnSe exhibits anisotropic thermoelectric transport properties due to its layered crystal structure and the uniaxial pressure of SPS process. Therefore, both the thermal and electrical transport properties of all the samples were measured parallel to the pressing direction in this work.

Revision: “Both the thermal and electrical transport properties of all the samples were measured parallel to the pressing direction” On page 21.

Comment: 7. The English writing needs to be more concise for better readability.

Response:

Thank you for your suggestion. We improved the English writing in the revised version.

Reviewer: 3

Comment: The authors systematically present WCl_6 doped SnSe to enhance the carrier concentration and obtain a large Seebeck coefficient. Microstructural characterization is also carried out to elucidate the reason for the low lattice thermal conductivity. They enabled polycrystalline SnSe to achieve a larger ZT value of 2.2 at 773 K. The results are summarized in the following table. The experiments are informative and fulfilling but suffer from the following problems.

Response:

We appreciate the positive and encouraging comments from the reviewer. And we are also very thankful for comments on how to further improve our paper for Nature Communications audience. According to your suggestions, we have further revised the manuscript and provide a point-to-point response.

Comment: 1. There is at least one terminological error in the manuscript, for example, in the abstract, “ultralow lattice conductivity” should be “ultralow lattice thermal conductivity”. Please check the manuscript carefully.

Response: Thank you for your suggestion. We have changed it and check the manuscript carefully.

Comment: 2. No explanation is given as to why the Seebeck coefficient of $\text{SnSe}_{0.92}+0.04\text{WCl}_6$ decreases with increasing temperature while all other trends increase with increasing temperature in Figure 3B.

Response:

We thank you for the insightful comment. To elucidate the decline in Seebeck coefficient with increasing temperature in the elevated temperature range for the $\text{SnSe}_{0.92}+0.04\text{WCl}_6$ compound, we carried out the density of states (DOS) calculations of a high W concentration in SnSe (~5% W doping, denoted as $\text{Sn}_{0.946}\text{W}_{0.054}\text{Se}_{0.92}$ in Fig. R8). Our analysis revealed that at a lower W concentration (such as 3%), a limited number of discrete defect energy levels manifest within the band gap, which forms the resonant energy state with the SnSe matrix to enhance the Seebeck coefficient (as depicted Fig. 4F in the manuscript). These discrete defect energy levels within the gap do not contribute to the electron transport, and the band gap remains akin to that of the pristine SnSe. However, at higher W concentrations (>4%), certain discrete defect energy levels within the band gap coalesce into the energy band (from 0 eV to ~0.5 eV in Fig. R8). This energy band obviously involves in the carrier transport. Consequently, the valence band maximum (VBM) shifts to the position ~0.5 eV and the band gap contracts to ~0.3 eV. As a result, the substantial reduction in the band gap due to high W doping instigates a bipolar effect at elevated temperatures, leading to a decline in the Seebeck coefficient.

To clarify this point, we have added the followings in the revised manuscript and supporting information:

“The Seebeck coefficient of $\text{SnSe}_{0.92} + 0.04\text{WCl}_6$ decreases with temperature at the high temperature region, which is due to the high doping concentration induced band gap shrinking and strong bipolar effect (Detailed in in Fig. S3).” On page 9.

“To elucidate the decline in Seebeck coefficient with increasing temperature in the elevated temperature range for the $\text{SnSe}_{0.92}+0.04\text{WCl}_6$ compound, we carried out the density of states (DOS) calculations of a high W concentration in SnSe (~5% W doping, denoted as $\text{Sn}_{0.946}\text{W}_{0.054}\text{Se}_{0.92}$ in Fig. S3). Our analysis revealed that at a lower W concentration (such as 3%), a limited number of discrete defect energy levels manifest within the band gap. These discrete defect energy levels within the gap do not contribute to the electron transport, and the band gap remains akin to that of the pristine SnSe. However, at higher W concentrations (>4%), certain discrete defect energy levels within the band gap coalesce into the energy band (from 0 eV to ~0.5 eV in Fig. S3).

This energy band obviously involves in the carrier transport. Consequently, the valence band maximum (VBM) shifts to the position ~ 0.5 eV and the band gap contracts to ~ 0.3 eV. As a result, the substantial reduction in the band gap due to high W doping instigates a bipolar effect at elevated temperatures, leading to a decline in the Seebeck coefficient.”
On page 9 of Supporting Information.

Fig. R8 Calculated density of states of $\text{Sn}_{0.946}\text{W}_{0.054}\text{Se}_{0.92}$ (the blue line) and pristine SnSe (the black line). The dashed line represents the Fermi energy level.

Comment: 3. The trends of total thermal conductivity and lattice thermal conductivity in Fig. 5 are similar due to the low electronic thermal conductivity, and it is sufficient to use the same comparison chart when making the comparative illustration without repeating the explanation.

Response:

Thanks for your suggestions. We have removed Fig. 5B (Comparison of κ_T), and revised Fig. 5 in the revised manuscript (Fig. R9).

Fig. R9. The temperature dependence of (A) total thermal conductivity (κ_T), (B) lattice thermal conductivity (κ_L), (C) Comparison of κ_L , (D) Calculated lattice thermal conductivities of SnSe compounds based on the Callaway's model. The black, red, green, orange and blue lines represent the lattice thermal conductivities of pristine SnSe, V_{Se} (with 8% Se vacancy in SnSe), V_{Sn} (with 14% Sn vacancy in SnSe), $V_{\text{Sn}}+V_{\text{Se}}$ (with 8% Se and 14% Sn vacancies in SnSe) and $3\%\text{WCl}_6+V_{\text{Sn}}+V_{\text{Se}}$ (with 8% Se and 14% Sn vacancies plus 3% WCl_6 doping in SnSe), respectively.

We hope that the revised manuscript meets the standards for publication in **Nature Communications**. Thanks for your time and consideration.

Sincerely and best regards

Guodong Tang, Yongsheng Zhang, Haijun Wu
On behalf of all co-authors

Reference

1. Shi X. L. et al. Advanced Thermoelectric Design: From Materials and Structures to Devices. *Chem. Rev.* **120**, 7399-7515 (2020).
2. Lee Y. K. et al. Surface Oxide Removal for Polycrystalline SnSe Reveals Near-Single-Crystal Thermoelectric Performance. *Joule* **3**, 719-731 (2019).
3. Xiao Y. et al. Origin of low thermal conductivity in SnSe. *Phys. Rev. B* **94**, (2016).

Reviewers' comments:

Reviewer #1 (Remarks to the Author):

In their response, the authors state, 'As compared with the pristine sample, the SnSe_{0.92}+0.03WCl₆ sample exhibits a significantly enhanced Seebeck coefficient, which is important for achieving high thermoelectric performance in our samples.' However, at 300K, the power factor (PF) of SnSe_{0.92}+0.03WCl₆ is similar to the pristine sample, despite the enhanced Seebeck coefficient. Therefore, it appears that the high thermoelectric performance may not be solely due to the enhanced Seebeck coefficient.

Furthermore, Figure R1 supports the reviewer's opinion, as S^2 only increases by about twice from 300K to 800K, while the growth of PF during the same temperature range is greater than twofold. This suggests that the growth trend of PF is more closely related to electrical conductivity rather than the Seebeck coefficient.

Editorial note: In his/her Remarks to the Editor, Reviewer #1 clearly state that he/she cannot recommend this paper due to contradictions in the analysis.

Reviewer #2 (Remarks to the Author):

A lot of improvements have been made in the revised manuscript. I am just still a little confused about the thermal conductivity. In Figure 5, SnSe_{0.92} + 0.03WCl₆ exhibits the lowest lattice thermal conductivities among all investigated samples, even lower than that of SnSe_{0.92} + 0.04WCl₆. Why doesn't the SnSe_{0.92} + 0.04WCl₆ sample achieve the lowest lattice thermal conductivities? More explanation or evidence need to provide. The paper can be published in NC after the above confusion are addressed.

Reviewer #3 (Remarks to the Author):

In the revised manuscript, the authors have explained in detail the issues raised by the review and supported their arguments by providing additional data and analysis, but the following issues remain

1. I think it is not reasonable enough in explaining how the Seebeck coefficient varies with temperature, and the experimentally observed phenomenon can be explained by measuring the variation of its carrier concentration with temperature to provide a theoretical justification.
2. Carrier concentration and mobility change with temperature can be used to further explain the conductivity change with temperature.

March 7, 2024

Dear Editor,

Many thanks for your correspondence concerning the Reviewers' comments on our manuscript (NCOMMS-24-02607A-Z) entitled "Divacancy and Resonance Level Enables High Thermoelectric Performance in n-Type SnSe Polycrystals". We are grateful for the Reviewers' positive remarks and constructive suggestions for improving our manuscript. We would also like to express our appreciation for your valuable input. We have modified our manuscript accordingly and the point-by-point response to the Reviewers' comments is given below.

Reviewer: 1

Comment: 1. In their response, the authors state, 'As compared with the pristine sample, the $\text{SnSe}_{0.92}+0.03\text{WCl}_6$ sample exhibits a significantly enhanced Seebeck coefficient, which is important for achieving high thermoelectric performance in our samples.' However, at 300K, the power factor (PF) of $\text{SnSe}_{0.92}+0.03\text{WCl}_6$ is similar to the pristine sample, despite the enhanced Seebeck coefficient. Therefore, it appears that the high thermoelectric performance may not be solely due to the enhanced Seebeck coefficient. Furthermore, Figure R1 supports the reviewer's opinion, as S^2 only increases by about twice from 300K to 800K, while the growth of PF during the same temperature range is greater than twofold. This suggests that the growth trend of PF is more closely related to electrical conductivity rather than the Seebeck coefficient.

The authors assert that the high power factor is due to a large Seebeck coefficient.

Response:

In fact, we didn't mentioned that the improved power factor is solely attributed to the enhanced Seebeck coefficient in both the initial manuscript and the revised version. As a matter of fact, we have always attributed to the enhanced power factor to the synergistic effects of enhanced electrical conductivity (or enhanced carrier concentration) and improved Seebeck coefficient. The followings are quoted from our manuscript. (1) In the abstract: "The combination of the enhanced carrier concentration (achieved through WCl_6 doping and Se vacancies) and large Seebeck coefficient lead to a high power factor"; (2) On page 8: "As a result, the sharp increase in n leads to a significant increase of σ "; (3) On page 11: "All $\text{SnSe}_{0.92} + x \text{WCl}_6$ samples exhibit significantly enhanced PF due to the simultaneous enhancement of both σ and S "; (4) In the conclusion: "W doping introduces resonance levels within the conduction bands

and increase DOS in the electronic structure of SnSe, consequently boosting the Seebeck coefficient. Moreover, the optimization of carrier concentration through WCl_6 doping and Se vacancies, ensures superior electrical transport properties.”

In addition to the Seebeck coefficient, the contribution of electrical conductivity is undeniably crucial for the enhancement of power factor in our work. It is noteworthy that our samples exhibit an increased carrier concentration, attributed to the presence of WCl_6 and Se vacancies. Therefore, another role of “divacancy” is to enhance the carrier concentration. The increased carrier concentration leads to a significant increase of electrical conductivity. The role of electrical conductivity contributing to high thermoelectric performance has also been emphasized in the manuscript Title “Divacancy and Resonance Level Enables High Thermoelectric Performance in n-Type SnSe Polycrystals”. Therefore, the innovative incorporation of the role of electrical conductivity in achieving high performance has been integrated into the main text and title of the original manuscript.

Our conclusion emphasizes the contribution of electrical conductivity and increased Seebeck coefficient to high performance, which is confirmed by Reviewer#3’s comment in the first-round review: **“The authors systematically present WCl_6 doped SnSe to enhance the carrier concentration and obtain a large Seebeck coefficient.”**

Reviewer: 2

Comment 1:

A lot of improvements have been made in the revised manuscript. I am just still a little confused about the thermal conductivity. In Figure 5, $\text{SnSe}_{0.92} + 0.03\text{WCl}_6$ exhibits the lowest lattice thermal conductivities among all investigated samples, even lower than that of $\text{SnSe}_{0.92} + 0.04\text{WCl}_6$. Why doesn’t the $\text{SnSe}_{0.92} + 0.04\text{WCl}_6$ sample achieve the lowest lattice thermal conductivities? More explanation or evidence need to provide. The paper can be published in NC after the above confusion are addressed.

Response:

We express our appreciation for the valuable suggestion provided. Upon doping 3% WCl_6 into $\text{SnSe}_{0.92}$, we observed the presence of W- and Cl-rich nanoprecipitates (Fig. 6 in the manuscript). This indicates that the solubility of WCl_6 is lower than 3%. Thus, the introduction of a higher concentration of WCl_6 (such as 4%) would result in

additional WCl_6 precipitates in the matrix. Given that the melting point of WCl_6 is lower than that of SnSe , the incorporation of WCl_6 as a sintering additive can expedite melting and lead to grain coarsening. To prove it, SEM analysis was conducted during the revision stage. As shown in Fig. R1 (Fig. S10), the grain sizes of $\text{SnSe}_{0.92} + 0.04\text{WCl}_6$ are much larger than those of $\text{SnSe}_{0.92} + 0.03\text{WCl}_6$. Based on the Debye-Callaway model, the $\tau_B^{-1} = \frac{v}{d}$, where τ_B represents the phonon relaxation time due to the grain boundaries, v is phonon velocity and d is the grain size. The larger grain size in the $\text{SnSe}_{0.92} + 0.04\text{WCl}_6$ matrix corresponds to the larger of the phonon relaxation time τ_B or the weaker boundary-phonon scattering effect. Therefore, $\text{SnSe}_{0.92} + 0.04\text{WCl}_6$ sample exhibits the higher lattice thermal conductivity than that of $\text{SnSe}_{0.92} + 0.03\text{WCl}_6$ due to grain coarsening.

Fig. R1 SEM images of (A) $\text{SnSe}_{0.92} + 0.03\text{WCl}_6$ and (B) $\text{SnSe}_{0.92} + 0.04\text{WCl}_6$ samples.

Revision: To clarify this point, we have added the followings in the revised manuscript: “As shown in Fig. 5B, $\text{SnSe}_{0.92} + 0.04\text{WCl}_6$ sample shows higher κ_L than that of $\text{SnSe}_{0.92} + 0.03\text{WCl}_6$. Upon doping 3% WCl_6 into $\text{SnSe}_{0.92}$, we observed the presence of W- and Cl-rich nanoprecipitates (Fig. 6). This indicates that the solubility of WCl_6 is lower than 3%. Thus, the introduction of a higher concentration of WCl_6 (such as 4%) would result in additional WCl_6 precipitates in the matrix. Given that the melting point of WCl_6 is lower than that of SnSe , the incorporation of WCl_6 as a sintering additive can expedite melting and lead to grain coarsening. As shown in Fig. S10, the grain sizes of $\text{SnSe}_{0.92} + 0.04\text{WCl}_6$ are much larger than those of $\text{SnSe}_{0.92} + 0.03\text{WCl}_6$. Based on the Debye-Callaway model, the $\tau_B^{-1} = \frac{v}{d}$, the larger grain size in the $\text{SnSe}_{0.92} + 0.04\text{WCl}_6$ matrix corresponds to the larger of the phonon relaxation time τ_B or the weaker boundary-phonon scattering effect. Therefore, $\text{SnSe}_{0.92} + 0.04\text{WCl}_6$ sample exhibits the higher lattice thermal conductivity than that of $\text{SnSe}_{0.92} + 0.03\text{WCl}_6$ due to grain coarsening.”

Reviewer: 3

Comment: In the revised manuscript, the authors have explained in detail the issues raised by the review and supported their arguments by providing additional data and analysis, but the following issues remain.

Response:

We appreciate the positive and encouraging comments from the reviewer. And we are also very thankful for comments on how to further improve our paper for Nature Communications audience. According to your suggestions, we have further revised the manuscript and provide a point-to-point response.

Comment: 1. I think it is not reasonable enough in explaining how the Seebeck coefficient varies with temperature, and the experimentally observed phenomenon can be explained by measuring the variation of its carrier concentration with temperature to provide a theoretical justification.

Response:

Following your suggestion, we have measured the variation of the carrier concentration (n) with temperature for the samples of 0.03 WCl₆ and 0.04WCl₆ (Fig. R2). If bipolar conduction occurs, carrier concentration (n) will quickly increase with temperature.^{1,2} Here, the carrier concentration slowly increases with temperature for the SnSe_{0.92} + 0.03WCl₆ sample below 600 K, which is similar with previously reported pure SnSe.³ Carrier concentration is stable with increasing temperature, indicating the absence of bipolar conduction. We found that n of SnSe_{0.92} + 0.04WCl₆ sample sharply increases with temperature from 510 K to 823 K. n of SnSe_{0.92} + 0.04WCl₆ reaches as high as $1.25 \times 10^{19} \text{ cm}^{-3}$ at 823 K, which is about two orders of magnitude larger than that of $1.3 \times 10^{17} \text{ cm}^{-3}$ at 510 K. The sharply increased carrier concentration suggests the onset of bipolar conduction for the SnSe_{0.92} + 0.04WCl₆ sample.^{1,2} The bipolar behavior is consistent with our band structure calculations (Fig. S4), which indicates the substantial reduction in the band gap due to high W doping. For a system with two types of carriers, the Seebeck coefficient can be described by²

$$S_{\text{total}} = \frac{\sigma_e S_e + \sigma_h S_h}{\sigma_{\text{total}}}$$

If the bipolar conduction occurs, the number of minorities (electrons) will increase obviously and the total S will reduce significantly. The temperature dependence of n is consistent with the trend in the Seebeck coefficient. Especially, with sharply increased carrier concentration from around 510 K, a decreased Seebeck coefficient can be

observed with rising temperature. Therefore, the bipolar conduction process leads to the gradual degraded S at elevated temperature in $\text{SnSe}_{0.92} + 0.04\text{WCl}_6$ sample.

Fig. R2 Temperature dependence of (a) carrier concentration (n), (b) carrier mobility (μ).

Revision: “The Seebeck coefficient of $\text{SnSe}_{0.92} + 0.04\text{WCl}_6$ decreases with increasing temperature, especially at the high-temperature regions. This is consistent with the sharply increased carrier concentration (Fig. S2), suggesting bipolar conduction. n of $\text{SnSe}_{0.92} + 0.04\text{WCl}_6$ reaches as high as $1.25 \times 10^{19} \text{cm}^{-3}$ at 823 K, which is about two orders of magnitude larger than that of $1.3 \times 10^{17} \text{cm}^{-3}$ at 510 K. The bipolar behavior is consistent with our band structure calculations (Detailed in Fig. S4), which indicates the substantial reduction in the band gap due to high W doping. Therefore, a bipolar conduction process is expected with rising temperature.”

Comment: 2. Carrier concentration and mobility change with temperature can be used to further explain the conductivity change with temperature.

Response:

Thanks for your suggestion. High-temperature Hall measurements were performed to further investigate the conductivity change with temperature. The temperature dependence of carrier concentration (n) and carrier mobility (μ) are shown in Fig. R2 and Fig. S2. The sharp increase of electrical conductivity at an intermediate temperature in our $\text{SnSe}_{0.92} - \text{WCl}_6$ samples can be attributed to both the increased carrier concentration and carrier mobility with temperature. $\text{SnSe}_{0.92} + 0.04\text{WCl}_6$ sample shows higher carrier mobility than that of $\text{SnSe}_{0.92} + 0.03\text{WCl}_6$ due to its larger grains (Fig. R1). The large grain size induces weak electron scattering at the grain boundary and higher carrier mobility.

Revision: “The temperature dependence of carrier concentration (n) and carrier

mobility (μ) were measured for further understanding the the electrical conductivity change with temperature (Fig. S2). The sharp increase of electrical conductivity at intermediate temperature can be attributed to both the increased carrier concentration and carrier mobility with temperature”

We hope that the revised manuscript meets the standards for publication in **Nature Communications**. Thanks for your time and consideration.

Sincerely and best regards

Guodong Tang, Yongsheng Zhang, Haijun Wu
On behalf of all co-authors

Reference

1. Hao F. et al. Enhanced Thermoelectric Performance in n-Type Bi₂Te₃-Based Alloys via Suppressing Intrinsic Excitation. *ACS. Appl. Mater. Inter.* **10**, 21372-21380 (2018).
2. Xing T. et al. Suppressed intrinsic excitation and enhanced thermoelectric performance in Ag_xBi_{0.5}Sb_{1.5-x}Te₃. *J Mater Chem C* **5**, 12619-12628 (2017).
3. Wei W. et al. Achieving high thermoelectric figure of merit in polycrystalline SnSe via introducing Sn vacancies. *J. Am. Chem. Soc.* **140**, 499 (2018)

REVIEWER COMMENTS

Reviewer #1 (Remarks to the Author):

(1) The analysis of the whole DFT focuses primarily on explaining the high Seebeck coefficient by highlighting the increased Density of States (DOS). Apart from DOS, other factors are considered simple and insufficient to explain the electrical transport properties comprehensively. The authors are encouraged to delve into a more physical discussion of these factors, rather than simply presenting the trend of band gap and energy levels. This could involve providing a deeper analysis of the implications of these trends on the material's electronic structure and its impact on the observed physical properties.

(2) The primary explanation for Power Factor (PF) can be simplified as follows: "Enhanced carrier concentration leads to increased electrical conductivity," and "Increased DOS results in a higher Seebeck coefficient." The reviewer recommends that authors pay more attention to the relationship between increased electrical conductivity and temperature variation since the growth of PF with temperature mirrors changes in electrical conductivity.

Reviewer #2 (Remarks to the Author):

The paper satisfactorily responds to my concerns, and the paper can be published in NC

Reviewer #3 (Remarks to the Author):

The revised manuscript and supporting documents show that the authors have taken our comments into account and revised them carefully. Here I think it could be published in the journal NATURE COMMUNICATION.

April 8, 2024

Many thanks for your correspondence concerning the Reviewers' comments on our manuscript (NCOMMS-24-02607B-Z) entitled "Divacancy and Resonance Level Enables High Thermoelectric Performance in n-Type SnSe Polycrystals". We are grateful for the Reviewers' positive remarks and constructive suggestions for improving our manuscript. We would also like to express our appreciation for your valuable input. We have modified our manuscript accordingly and the point-by-point response to the Reviewers' comments is given below.

Reviewer: 1

Comment: (1) The analysis of the whole DFT focuses primarily on explaining the high Seebeck coefficient by highlighting the increased Density of States (DOS). Apart from DOS, other factors are considered simple and insufficient to explain the electrical transport properties comprehensively. The authors are encouraged to delve into a more physical discussion of these factors, rather than simply presenting the trend of band gap and energy levels. This could involve providing a deeper analysis of the implications of these trends on the material's electronic structure and its impact on the observed physical properties.

Response:

Thanks for your suggestions. We agree with you that other factors should be considered to deeply understand the physical properties of the compounds. Due to the layered structure of the pristine SnSe compound, the interlayer electrical conductivity is very low.¹⁻³ Thus, the total electrical conductivity of pristine SnSe polycrystals is also low (Fig. 3a). Interestingly, from theoretical simulations of W dopant in SnSe, our investigation reveals that the doping W forms the W-Se bonds not only within the SnSe layer but also with the SnSe interlayers. This means that the W atom bridges two Sn-Se layers (Fig. 4d). The interlayer bonds will facilitate the electron transport across the layers, and boost the electrical conductivity. The formation of W-Se bonds indicates that the strong interactions between W and the SnSe matrix. This will manipulate the electronic structure of the SnSe matrix, such as inducing the resonant energy level around the conduction band minimum (CBM) and more density of states (DOS) peaks below the Fermi level (Figs. 4e, 4f). The heightened resonant energy levels contribute to the enhancement of the Seebeck coefficient. Moreover, the introduction of W dopant leads to the emergence of additional DOS peaks below the Fermi level means that W doped SnSe_{0.92} has a higher electron carrier concentration than SnSe_{0.92}. This is

consistent with the experimentally observed rise in carrier concentration and electrical conductivity of W doping compared with the pristine SnSe_{0.92} (Table 1 and Fig. 3a).

To clarify this point, we have added the followings in the revised manuscript.

“To elucidate the origin of the enhanced electrical transport properties”

“For the pristine SnSe compound, due to the layered structure, the interlayer electrical conductivity is very low.^{1, 20, 21} Thus, the total electrical conductivity of pristine SnSe polycrystals is also low (Fig. 3a). However, the W dopant in SnSe can bridge two Sn-Se layers (Fig. 4d). The interlayer bonds will facilitate the electron transport across the layers, and boost the electrical conductivity. This is consistent with the experimentally measured increased electrical conductivity of W doping compared to those of the pristine SnSe (Fig. 3a).”

“Unlike the defect energy level in the DOS of SnSe_{0.92}, the formation of W-Se bonds in W doped SnSe indicates that the strong interactions between W and the SnSe matrix. This will modify the corresponding electronic structures of SnSe matrix and form the resonant energy level just below the conduction band minimum (CBM) and more DOS peaks below the Fermi level (Figs. 4e, 4f). Actually, this resonant DOS peak is from the d energy level of W dopant (Supplementary Fig. 7c)”

“Moreover, the introduction of W dopant leads to the emergence of additional DOS peaks below the Fermi level means that W doped SnSe_{0.92} has a higher electron carrier concentration than SnSe_{0.92}. The increased DOS (or the induced resonant energy level around the CBM), the W-Se bonds bridging SnSe interlayers and the increased carrier concentration will significantly boost the Seebeck coefficient and electrical conductivity of W doped SnSe_{0.92}, respectively. These are consistent with the experimental observations.”

Comment: (2) The primary explanation for Power Factor (PF) can be simplified as follows: "Enhanced carrier concentration leads to increased electrical conductivity," and "Increased DOS results in a higher Seebeck coefficient." The reviewer recommends that authors pay more attention to the relationship between increased electrical conductivity and temperature variation since the growth of PF with temperature mirrors changes in electrical conductivity.

Response:

Thanks for your suggestions. We add more discussion about relationship between increased electrical conductivity and temperature variation in the revised manuscript.

To clarify this point, we have added the followings in the revised manuscript.

“ σ of high performance SnSe_{0.92} + 3% WCl₆ sample maintain very low value from

300 to 475 K. Then σ sharply increases with increasing temperature above 475 K.”
“The carrier concentration of the $\text{SnSe}_{0.92} + 0.03\text{WCl}_6$ sample increase above 475 K, which resulting from thermal activation.³¹ In the meanwhile, the carrier mobility abruptly increases with temperature. Therefore, the notable increase of σ above 475 K can be ascribed to both the increased carrier concentration and carrier mobility.”
“W doping and Se vacancies enhance carrier concentration and electrical conductivity in the whole temperature range. It is worth to note that the electrical conductivity abruptly increases with temperature above 475 K due to the largely increased carrier concentration and carrier mobility. W doping improves Seebeck coefficient through creating resonant energy level. These synergistic effects help to achieve significantly enhanced *PF*.”

Reviewer:2

Comment: The paper satisfactorily responds to my concerns, and the paper can be published in NC.

Response:

We appreciate the positive comments from the reviewer.

Reviewer:3

Comment: The revised manuscript and supporting documents show that the authors have taken our comments into account and revised them carefully. Here I think it could be published in the journal NATURE COMMUNICATION.

Response:

We appreciate the positive comments from the reviewer.

We hope that the revised manuscript meets the standards for publication in **Nature Communications**. Thanks for your time and consideration.

Sincerely and best regards

Guodong Tang, Yongsheng Zhang, Haijun Wu
On behalf of all co-authors

Reference

1. Shi X.-L. et al. Rational Structural Design and Manipulation Advance SnSe Thermoelectrics. *Mater. Horizons* **7**, 3065-3096 (2020).
2. Shi X. L. et al. High-Performance Thermoelectric SnSe: Aqueous Synthesis, Innovations, and Challenges. *Adv. Sci.* **7**, 1902923 (2020).
3. Zhao L.-D. et al. Ultralow Thermal Conductivity and High Thermoelectric Figure of Merit in SnSe Crystals. *Nature* **508**, 373-377 (2014).

REVIEWERS' COMMENTS

Reviewer #1 (Remarks to the Author):

The updated manuscript and accompanying materials demonstrate that the authors have thoughtfully addressed feedback and made revisions accordingly. It is suitable for publication in the Nature Communications.

April 20, 2024

Many thanks for your correspondence concerning the Reviewers' comments on our manuscript (NCOMMS-24-02607C) entitled "Divacancy and Resonance Level Enables High Thermoelectric Performance in n-Type SnSe Polycrystals". We are grateful for the Reviewers' positive comment for our manuscript.

Reviewer: 1

Comment: The updated manuscript and accompanying materials demonstrate that the authors have thoughtfully addressed feedback and made revisions accordingly. It is suitable for publication in the Nature Communications.

Response:

We appreciate the positive comments from the reviewer.

We hope that the revised manuscript meets the standards for publication in **Nature Communications**. Thanks for your time and consideration.

Sincerely and best regards

Guodong Tang, Yongsheng Zhang, Haijun Wu
On behalf of all co-authors